# Learning-Augmented Algorithms for MTS with Bandit Access to Multiple Predictors

**Matei Gabriel Coşa** [1]    **Marek Eliáš** [1]

## Abstract

We consider the following problem: we are given $\ell$ heuristics for Metrical Task Systems (MTS), where each might be tailored to a different type of input instances. While processing an input instance received online, we are allowed to query the action of only one of the heuristics at each time step. Our goal is to achieve performance comparable to the best of the given heuristics. The main difficulty of our setting comes from the fact that the cost paid by a heuristic at time $t$ cannot be estimated unless the same heuristic was also queried at time $t-1$. This is related to Bandit Learning against memory bounded adversaries (Arora et al., 2012). We show how to achieve regret of $O(\mathrm{OPT}^{2/3})$ and prove a tight lower bound based on the construction of Dekel et al. (2013).

## 1. Introduction

*Metrical Task Systems (MTS)* (Borodin et al., 1992) are a very broad class of online problems capable of modeling problems arising in computing, production systems, power management, and routing service vehicles. In fact, many fundamental problems in the field of Online Algorithms including Caching, $k$-server, Ski-rental, and Convex Body Chasing are special cases of MTS. Metrical Task Systems are also related to *Online Learning from Expert Advice*, see (Blum & Burch, 2000) for the comparison of the two problems.

In MTS, we are given a description of a metric space $(M, d)$ and a starting point $s_0 \in M$ beforehand. The points in $M$ are traditionally called *states* and, depending on the setting, they can represent actions, investment strategies, or configurations of some complex system. At each time step $t = 1, \ldots, T$, we receive a cost function $c_t \colon M \to \mathbb{R}_+ \cup \{0, +\infty\}$. Seeing $c_t$, we can decide to stay in our previous state $s_{t-1}$ and pay its cost $c_t(s_{t-1})$, or move to some (possibly cheaper) state $s_t$ and pay $c_t(s_t) + d(s_{t-1}, s_t)$, where the distance function $d$ represents the transition cost between two states. Our objective is to minimize the total cost paid over time.

MTS is very hard in the worst case. This is due to its online nature ($s_t$ has to be chosen without knowledge of $c_{t+1}, \ldots, c_T$) and also due to its generality. The performance of algorithms is evaluated using the competitive ratio which is, roughly speaking, the ratio between the algorithm's cost and the cost of the optimal solution computed offline. Denoting $n$ the number of points in the metric space $M$, the best competitive ratio achievable for MTS is $2n - 1$ and $\Theta(\log^2 n)$ in the case of deterministic and randomized algorithms, respectively (Borodin et al., 1992; Bartal et al., 2006; Bubeck et al., 2022a; 2019). Note that $n$ is usually very large or even infinite (e.g., $M = \mathbb{R}^d$).

This worst-case hardness motivates the study of MTS and its special cases in the context of *Learning-Augmented Algorithms* (Lykouris & Vassilvitskii, 2021). Here, the algorithm can use predictions produced by an ML model in order to exploit specific properties of input instances. By augmenting the algorithm with the ML model, we obtain a heuristic with an outstanding performance, well beyond the classical worst-case lower bounds, on all inputs where the ML model performs well. One of the key techniques used in this context is *Combining Heuristics* which is used to achieve robustness (Lykouris & Vassilvitskii, 2021; Wei, 2020; Antoniadis et al., 2023a), adjust hyperparameters (Antoniadis et al., 2021), and recognize the most suitable heuristic for the current input instance (Emek et al., 2021; Anand et al., 2022; Antoniadis et al., 2023b).

**Combining heuristics.** In this basic theoretical problem, we are given $\ell$ heuristics $H_1, \ldots, H_\ell$ which can be simulated on the input instance received online. We want to combine them into a single algorithm which achieves a cost comparable to the best of the heuristics on each individual instance. The difficulty of the combination problem is given by the online setting: because we receive the input online,

[1]Department of Computing Sciences, Bocconi University, Milan, Italy. Correspondence to: Matei Gabriel Coşa <mateigabriel.cosa@studbocconi.it>, Marek Eliáš <marek.elias@unibocconi.it>.

*Proceedings of the $42^{nd}$ International Conference on Machine Learning*, Vancouver, Canada. PMLR 267, 2025. Copyright 2025 by the author(s).

we cannot tell beforehand which heuristic is going to perform better. Moreover, steps performed while following a wrong heuristic cannot be revoked.

We can combine a heuristic deploying an ML prediction model with a classical online algorithm in order to obtain a new algorithm that satisfies worst-case guarantees regardless of the performance of the ML model (this property is called *robustness*). Similarly, we can use it to deploy a portfolio of very specialized ML models in a broader setting, where each model corresponds to a separate heuristic. The combination technique ensures that we achieve very good performance whenever at least one of the models performs well on the current input instance.

**Full-feedback setting.** Most of the previous works on Combining Heuristics query the state of each heuristic at each time step, observe their costs, and choose between their states, see (Wei, 2020; Antoniadis et al., 2023a; Blum & Burch, 2000; Fiat et al., 1991). We call this the *full-feedback setting* and, indeed, methods from Online Learning in the full-feedback setting can be directly integrated in the combination algorithms. For example, Blum & Burch (2000) showed that by choosing between heuristics using the HEDGE algorithm (Freund & Schapire (1997)) with a well-chosen learning rate we can be $(1 + \epsilon)$-competitive with respect to the best heuristic.[1]

**Bandit-feedback setting.** In this paper, we consider what can be seen as a bandit-feedback setting of Combining Heuristics. At each time step, we are allowed to query the current state of only one heuristic. This setting allows us to study the impact of restricting information about the heuristics on our ability to combine them effectively. Further motivation is the fact that querying all the heuristics is costly, especially if they utilize some heavy-weight prediction models. This issue was already considered by Emek et al. (2021) and Antoniadis et al. (2023b) whose works inspired our study. Note that the combination algorithm retains full access to the input instance. This is required by any positive result for general MTS, since the cost functions $c_t$ usually do not satisfy any natural properties such as Lipschitzness or convexity; see further discussion in Section 2. Moreover, $c_t$ is often easy to encode. For example, in Caching, $c_t$ is fully determined by the page requested at time $t$.

In contrast to the full-feedback setting, classical learning methods cannot be directly integrated in combination algorithms operating in the bandit setting. The main reason is that the state $s_t^i$ of the heuristic $H_i$ is not enough to estimate its cost at time $t$: the cost paid by $H_i$ at time $t$ is

$c_t(s_t^i) + d(s_{t-1}^i, s_t^i)$, i.e., we cannot calculate this cost[2] unless we have queried $H_i$ in both time steps $t - 1$ and $t$. We also consider the case where each heuristic needs to be *bootstrapped* for $m - 2 \geq 0$ time steps before we can see its state. This way, to receive the state of $H_i$ in time steps $t - 1$ and $t$, and to be able to calculate its cost at time $t$, we have to query $H_i$ in the $m$ time steps $t - m + 1, \ldots, t$.

A similar phenomenon occurs in the setting of *Online Learning against Memory-Bounded Adversaries* (Arora et al., 2012), which is our main theoretical motivation for considering non-zero bootsrapping time (i.e., $m > 2$). There, one needs to play an action $a$ at least $m$ times in a row to see the loss of the reference policy which plays $a$ at each time step. The algorithm of Arora et al. (2012) splits the time horizon into blocks of length larger than $m$. In the block $i$, they keep playing the same action $a_i$ which allows them to observe the loss of the corresponding reference policy after the first $m$ time steps in the block. However, this does not work in our setting. The adversary can set up the MTS instance in such a way that $c_t \neq 0$ and the heuristics move only on the boundaries of the blocks, so that the algorithm would not observe the cost of any heuristic. This way, the algorithm cannot be competitive with respect to the best heuristic.

A different bandit-like setting for Combining Heuristics for MTS was proposed by Antoniadis et al. (2023b). In their setting, the queried predictor reports both its state and the declared moving cost incurred at time $t$. We discuss their contribution in Section 1.2. Their $(1 + \epsilon)$-competitive algorithm and absence of lower bounds motivated us to study regret bounds which can provide more refined guarantees when the competitive ratio is close to 1. However, we have decided to pursue these bounds in the more natural setting where the moving costs are not reported as it does not rely on the honesty of the predictors.[3] Since our algorithms use a subset of information available in the setting of Antoniadis et al. (2023b), our upper bounds also apply to their setting.

## 1.1. Our Results

For $m \geq 2$, we say that an algorithm ALG has $m$-*delayed bandit access* to heuristics $H_1, \ldots, H_\ell$ if (i) it can query at most one heuristic $H_i$ at each time step $t$ and (ii) the query yields the current state of $H_i$ only if $H_i$ was queried also in steps $t - m + 2, \ldots, t$, otherwise it yields an empty result.

---

[1]In fact, one can achieve cost $H^* + O(D\sqrt{H^*})$, where $H^*$ denotes the cost of the best heuristic and $D$ the diameter of the metric space $M$.

[2]In fact, we cannot even estimate it. In Caching, $k$-server, and Convex Body Chasing, to give a few examples, the cost of each state is either 0 or $+\infty$. Therefore, any algorithm with a finite cost is always located at some state $s_t$ such that $c_t(s_t) = 0$ and the difference in the performance of the algorithms can be seen only in the moving costs.

[3]Alternatively, we could require predictors to certify their moving cost by providing their state in the previous time step. This way, however, we end up querying the state of two predictors in a single time step.

Our main result is an algorithm with the following regret.

**Theorem 1.1.** *Let $D$, $\ell$, and $m \geq 2$ be constant. Consider heuristics $H_1, \ldots, H_\ell$ for an MTS with diameter $D$. There is an algorithm* ALG *which, given an $m$-delayed bandit access to $H_1, \ldots, H_\ell$, satisfies the following guarantee. The expected cost of* ALG *on any input $I$ is*

$$\mathbb{E}[\text{ALG}] \leq \text{OPT}_{\leq 0} + O\big(\text{OPT}_{\leq 0}^{2/3}\big),$$

*where $\text{OPT}_{\leq 0} = \min_{i=1}^{\ell} H(I)$ denotes the cost of the best heuristic on input $I$.*

We prove this result in Section 3, where we state explicitly the dependence on $D$, $m$, and $\ell$ in the case when they are larger than a constant. We can interpret our result in terms of the competitive ratio. Since the regret term in Theorem 1.1 is sublinear in $\text{OPT}_{\leq 0}$, the expected cost of our algorithm is at most $\mathbb{E}[\text{ALG}] \leq (1 + o(1)) \, \text{OPT}_{\leq 0}$. In other words, its competitive ratio with respect to the best of the $\ell$ heuristics converges to 1. Therefore, our algorithm can be used for robustification. If we include a classical online algorithm which is $\rho$-competitive in the worst case among $H_1, \ldots, H_\ell$, our algorithm will never be worse than $(1 + o(1))\rho$-competitive. However, on input instances where some of the heuristics incur a very low cost, the algorithm can match their performance asymptotically. For example, if the cost of the best heuristic is only 1.01 times higher than the offline optimum, our algorithm's cost will be only $(1 + o(1)) \cdot 1.01$ times the offline optimum.

Our algorithm alternates between exploitation and exploration, as in the classical approach for the *Multi-Armed Bandit (MAB)* setting, see e.g. (Slivkins, 2019). However, each exploration phase takes $m$ time steps and we are unable to build an unbiased estimator of the loss vectors. Moreover, our algorithm sometimes needs to take improper steps, i.e., going to a state which is not suggested by any of the predictors. This is clearly necessary with $m > 2$, since we may receive empty answers to our queries. Such improper steps are crucial for achieving regret in terms of $\text{OPT}_{\leq 0}$ even for $m = 2$.

In Appendix D, we show that our algorithm can be adapted to the $m$-memory bounded setting for bandits, where it achieves a regret of $O(T^{2/3})$ comparable to Arora et al. (2012), while slightly improving the dependence on $m$. Note that $T$ can be much larger than $\text{OPT}_{\leq 0}$.

We extend our result to a setting with a benchmark which can switch between the heuristics at most $k$ times, while the algorithm's number of switches still remains unrestricted.

**Theorem 1.2.** *Let $D$, $\ell$, and $m \geq 2$ be constant and $k \geq 1$ be a parameter. Consider heuristics $H_1, \ldots, H_\ell$ for an MTS with diameter $D$. There is an algorithm* ALG *which, given an $m$-delayed bandit access to $H_1, \ldots, H_\ell$, satisfies*

*the following guarantee. The cost of* ALG *on input $I$ with offline optimum cost at least $2k$ is*

$$\mathbb{E}[\text{ALG}] \leq \text{OPT}_{\leq k} + \tilde{O}(k^{1/3} \, \text{OPT}_{\leq k}^{2/3}),$$

*where $\text{OPT}_{\leq k}$ denotes the cost of the best combination of heuristics in hindsight that switches between heuristics at most $k$ times on input $I$.*

Again, we can interpret this regret bound as a competitive ratio converging to 1. In Appendix C, we show that the competitive ratio of our algorithm is below $(1 + \epsilon)$ for $k$ as large as $\tilde{\Omega}(\epsilon^3 \, \text{OPT}_{\leq k})$ which is worse by a log-factor than the result of Antoniadis et al. (2023b) in their easier setting.

In Section 4, we show that our upper bound in Theorem 1.1 is tight up to a logarithmic factor even with 0 bootstrapping time ($m = 2$). Our result is based on the construction of Dekel et al. (2013) for *Bandits with Switching Costs*. Their lower bound cannot be applied directly, since algorithms in our setting have an advantage in being able to take improper actions and having one-step look-ahead. Both of these advantages come from the nature of MTS and are indispensable in our setting, see Section 2.

**Theorem 1.3.** *For any algorithm* ALG *with 2-delayed bandit access to $\ell$ predictors, there is an input instance $I$ such that the expected cost of* ALG *is*

$$\mathbb{E}[\text{ALG}] \geq \text{OPT}_{\leq 0} + \tilde{\Omega}(\text{OPT}_{\leq 0}^{2/3}),$$

*where $\text{OPT}_{\leq 0} = \min_{i=1}^{\ell} H(I)$ denotes the cost of the best heuristic on input $I$.*

In Appendix B.1, we also show that the dependence on $D, \ell$ and $k$ in our bounds is (almost) optimal. In particular, our regret bound in Theorem 1.1 scales with $(Dk\ell \ln \ell)^{1/3} m^{2/3}$ and we show that this dependence needs to be at least $(Dk\ell)^{1/3}$ with $m = 2$.

## 1.2. Related Work

Arora et al. (2012) introduced the problem of Bandit Learning against Memory-Bounded Adversaries. Here, the loss functions depend on the last $\mu + 1$ actions taken by the algorithm. This setting captures, for example, Bandits with Switching Costs (Amir et al., 2022; Rouyer et al., 2021). They propose an elegant algorithm with regret $O(\mu T^{2/3})$ that partitions the time horizon into blocks of equal size and let a classical online learning algorithm (e.g., EXP3) play over the losses aggregated in each block. Their result was shown to be tight by Dekel et al. (2013) who provided a sophisticated lower bound construction showing that any algorithm suffers regret at least $\tilde{\Omega}(T^{2/3})$ already for $\mu = 1$. Our results are also related to *Non-Stationary Bandits* and *Dynamic Regret* (Auer et al., 2002, Section 8).

Antoniadis et al. (2023b) studied dynamic combination of heuristics for MTS. By reducing to the *Layered Graph Traversal* problem (Bubeck et al., 2022b), they achieved a competitive ratio $O(\ell^2)$ with respect to the best dynamic combination of $\ell$ heuristics. Then, they focused on the scenario where the input instance is partitioned into $k$ intervals and a different heuristic excels in each of the intervals. They provided bounds on how big $k$ can be to make $(1 + \epsilon)$-competitive algorithms possible. Finally, they also studied this question in the bandit-like setting which is strictly easier compared to ours.

First works on Combining Heuristics were by Fiat et al. (1990) for $k$-server, Fiat et al. (1991) for Caching, and Azar et al. (1993); Blum & Burch (2000) for MTS. More recently, Emek et al. (2021) studied Caching with multiple predictors and achieved regret sublinear in $T$, while also tackling a bandit-like setting. Further results on other online problems are by Anand et al. (2022); Dinitz et al. (2022); Bhaskara et al. (2020); Gollapudi & Panigrahi (2019); Almanza et al. (2021); Wang et al. (2020); Kevi & Nguyen (2023).

Learning-augmented algorithms were introduced by Lykouris & Vassilvitskii (2021); Kraska et al. (2018) who designed algorithms effectively utilizing unreliable machine-learned predictions. Since these two seminal works, many computational problems were studied in this setting, including Caching (Rohatgi, 2020; Wei, 2020), Scheduling (Lindermayr & Megow, 2022; Benomar & Perchet, 2024b; Balkanski et al., 2023; Bamas et al., 2020), graph problems (Eberle et al., 2022; Bernardini et al., 2022; Dong et al., 2025; Davies et al., 2023) and others. Several works consider algorithms using the predictions sparingly (Im et al., 2022; Drygala et al., 2023; Sadek & Eliáš, 2024; Benomar & Perchet, 2024a). See the survey (Mitzenmacher & Vassilvitskii, 2022) and the website by (Lindermayr & Megow, 2023).

Metrical Task Systems were introduced by Borodin et al. (1992) who gave a tight competitive ratio of $2n - 1$ for deterministic algorithms ($n$ is the number of states). The best competitive ratio for general MTS is $\Theta(\log^2 n)$ by Bubeck et al. (2019) and Bubeck et al. (2022a).

## 2. Notation and Preliminaries

We consider MTS instances with a bounded diameter and denote $D = \max_{s,s' \in M} d(s, s')$ the diameter of the underlying metric space. For example, $D$ in caching is equal to the size of the cache. At each time step, the algorithm receives the cost function $c_t$ first, and then it chooses its new state $s_t$, i.e. there is a 1-step look-ahead. This is standard in MTS definition and it is necessary for existence of any competitive algorithm, since $c_t$ is potentially unbounded, see (Blum & Burch, 2000, Section 2.3). We denote $\Delta^d$ the

$d$-dimensional probability simplex, and $[d] = \{1, \ldots, d\}$.

$m$**-delayed bandit access to heuristics.** Given $\ell$ heuristics $H_1, \ldots, H_\ell$, we denote $s_t^i$ the state of $H_i$ at time $t$ and $f_t(i) = c_t(s_t^i) + d(s_{t-1}^i, s_t^i)$ the cost incurred by $H_i$ at time $t$. Let $m \geq 2$ be a parameter. At each time $t$, the algorithm is allowed to query a single heuristic $H_i$. If $H_i$ was also queried in time steps $t - m + 2, \ldots, t$, the result of the query is the state $s_t^i$. Otherwise, the result is empty. While the access to the states of the heuristics is restricted, the algorithm has full access to the input instance which is not related to acquiring costly predictive information. Moreover, the input instance can be often described in a very compact way. For example, $c_t$ in Caching is completely determined by the page requested at time $t$. Access to the input instance is necessary because the cost functions are not required to satisfy any natural assumptions (like boundedness, Lipschitzness, convexity). For example, if the queried heuristic reports a state $s$ with $c_t(s) = +\infty$, the algorithm needs to know $c_t$ in order to choose a different state and avoid paying the infinite cost. Note that a similar situation can easily happen in Caching, $k$-server, Convex Body Chasing, or Convex Function Chasing.

We assume that $f_t(i) \in [0, 2D]$. This is without loss of generality for the following reason. First, we can assume that at each time $t$ there is a state with zero cost, since subtracting $\min_s c_t(s)$ from the cost of each state affects the cost of any algorithm (including the offline optimum) equally. Second, any predictor can be post-processed so that, in each time step where its cost is higher than $2D$, it serves the request in the state with $0$ cost and returns to the predicted state, paying at most $2D$ for the movement.

**Benchmarks and performance metrics.** Let $\text{OPT}_{\leq 0} = \min_{i=1}^{\ell} \sum_{t=1}^{T} f_t(i)$ be the static optimum, i.e., the cost of the best heuristic. For $k > 0$, we define

$$\text{OPT}_{\leq k} = \min_{i_1, \ldots, i_T} \sum_{t=1}^{T} \left( c_t(s_t^{i_t}) + d(s_{t-1}^{i_{t-1}}, s_t^{i_t}) \right)$$
$$\geq \min_{i_1, \ldots, i_T} \sum_{t=1}^{T} f_t(i_t) - kD,$$

where the minimum is taken over all solutions $i_1, \ldots, i_T$ such that the number of steps where $i_{t-1} \neq i_t$ is at most $k$.

We evaluate the performance of our algorithms using *expected pseudoregret* regret (further abbreviated as *regret*). For $k \geq 0$, we define

$$\text{Reg}_k(\text{ALG}) = \mathbb{E}[\text{ALG}] - \text{OPT}_{\leq k},$$

where ALG denotes the cost incurred by the algorithm on the given input instance with access to the given heuristics. We assume that the adversary is *oblivious* and has to fix the MTS input instance and the solutions of the heuristics before seeing the random bits of the algorithm. We say

that an algorithm is $\rho$-competitive with respect to an offline algorithm OFF, if $\mathbb{E}[\text{ALG}] \leq \rho \, \text{OFF} + \alpha$ holds on every input instances, where $\alpha$ is a constant independent on the input instance and we use OFF to denote both the algorithm and its cost. If OFF is an offline optimal algorithm, we call $\rho$ the competitive ratio of ALG.

**Rounding fractional algorithms.** A fractional algorithm for MTS is an algorithm which, at each time $t$, produces a distribution $p_t \in \Delta^M$ over the states in $M$. These distributions do not yet say much about the movement costs of the algorithm. But there is a standard way to turn such a fractional algorithm into a randomized algorithm for MTS.

**Proposition 2.1.** *There is an online randomized algorithm for MTS which, receiving online a sequence of distributions $p_1, \ldots, p_T \in \Delta^M$, produces a solution $s_1, \ldots, s_T \in M$ with expected cost equal to*

$$\mathbb{E}[\text{ALG}] = \sum_{t=1}^{T} \left( c_t^T p_t + \text{EMD}(p_{t-1}, p_t) \right),$$

*where* EMD *denotes the Earth mover distance with respect to the metric space $M$.*

We include the proof of this standard fact in Appendix E together with the following, very similar, proposition, where we overestimated the cost of switching between the states of two heuristics by $D$.

**Proposition 2.2.** *There is an online randomized algorithm which, receiving online a sequence of distributions $x_1, \ldots, x_T \in \Delta^\ell$ over the heuristics, queries at each time $t$ a heuristic $i_t$ with probability $x_t(i_t)$ such that*

$$\mathbb{E}\left[ \sum_{t=1}^{T} c_t(s_t^{i_t}) + d(s_{t-1}^{i_{t-1}}, s_t^{i_t}) \right] \leq \sum_{t=1}^{T} f_t^T x_t + \frac{D}{2} \|x_{t-1} - x_t\|_1.$$

**Basic learning algorithms.** We use the classical algorithms for online learning with expert advice HEDGE (Freund & Schapire (1997)) and SHARE (Herbster & Warmuth (1998)). Both satisfy the following property with $\eta$ being their learning rate. For both of them, the proof is contained in (Blum & Burch, 2000), we discuss more details, as well as the learning dynamics in Appendix F.

**Property 2.3.** There is a parameter $\eta$ such that the following holds. Denoting $x_{t-1}$ and $x_t$ the solutions of the algorithm before and after receiving loss vector $g_{t-1}$, we have

$$\|x_{t-1} - x_t\|_1 \leq \eta g_{t-1}^T x_{t-1}.$$

We use the bounds for HEDGE tuned for "small losses", see (Cesa-Bianchi & Lugosi, 2006).

**Proposition 2.4.** *Consider $x_1, \ldots, x_T \in \Delta^\ell$ the solution produced by HEDGE with learning rate $\eta$ and denote $\gamma := 1 - \exp(-\eta)$. For any $x^* \in \Delta^\ell$, we have*

$$\sum_{t=1}^{T} g_t^T x_t \leq \frac{\eta \sum_{t=1}^{T} g_t^T x^* + \ln \ell}{1 - \exp(-\eta)} \leq (1+\gamma) \sum_{t=1}^{T} g_t^T x^* + \frac{\ln \ell}{\gamma}.$$

## 3. Algorithm for $m$-Delayed Bandit Access to Heuristics

### 3.1. Basic Approach and Comparison to Previous Works

Arora et al. (2012) use the following approach to limit the number of switches between arms (or heuristics): split the time horizon into blocks of length $\tau$ and use some MAB algorithm to choose a single arm (or heuristic) for each block which is then played during the whole block. The number of switches is then at most $T/\tau$. This is a common approach to reduce the number of switches, see (Rouyer et al., 2021; Amir et al., 2022; Blum & Mansour, 2007). However, this approach does not lead to regret sublinear in $\text{OPT}_{\leq 0}$ which can be much smaller than $T$. In order to have the number of switches $T/\tau \leq o(\text{OPT}_{\leq 0})$, we have to choose $\tau = \omega(T/\text{OPT}_{\leq 0})$ which can be $\omega(\text{OPT}_{\leq 0})$ for small $\text{OPT}_{\leq 0}$. However, with blocks so large, already a single exploration of some arbitrarily bad heuristic would induce a cost of order $\tau \geq \omega(\text{OPT}_{\leq 0})$.

In turn, our algorithm is more similar to the classical MAB algorithm alternating exploration and exploitation steps, see (Slivkins, 2019). However, there are three key differences and each of them is necessary to achieve our performance guarantees:

- Our algorithm makes improper steps (i.e., steps not taken by any of the heuristics);

- We use MTS-style rounding to ensure bounded switching cost instead of independent random choice at each time step;

- Exploration steps are not sampled independently since our setting requires $m \geq 2$.

In particular, the last difference leads to more involving analysis. This is because we cannot assume that we have an unbiased estimator of the loss vector and consequently need to do extensive conditioning on past events. Moreover, the cost of only one of the time steps during each exploration phase can be directly charged to the expected loss of the internal full-feedback algorithm. We need to exploit the stability property of the internal full-feedback algorithm in order to relate the costs incurred during the steps of each exploration block.

During each exploitation step $t$, our algorithm follows the advice of the *exploited* heuristic which is sampled from the algorithm's internal distribution $x_t$ over the heuristics. Each exploration step $t$ is set up so that the algorithm discovers the cost of the heuristic $H_{e_t}$ chosen uniformly at random and updates its distribution over the heuristics. However, the algorithm does not follow $H_{e_t}$. Instead, it makes a greedy step from the last known state of the exploited heuristic.

## 3.2. Description

Let $\bar{A}$ be an algorithm for the classical learning from expert advice in full feedback setting (e.g. HEDGE or SHARE), and $\epsilon$ be a parameter controlling our exploration rate. For each time step $t = 1, \ldots, T$, we sample $\beta_t \in \{0, 1\}$ such that $\beta_t = 1$ with probability $\epsilon$ and $e_t$ is chosen uniformly at random from $\{1, \ldots, \ell\}$. We set $\beta_0 = 0$ since the algorithm starts querying predictors from $t = 1$. Moreover, we assume all heuristics reside in $s_0$ at $t = 0$. If $t$ is an exploitation step ($t \in X$) and $\beta_t = 0$, the next step will be again exploitation. Otherwise, the algorithm skips $m$ time steps which are needed to bootstrap the explored heuristic $H_{e_{t+m}}$ and performs exploration in step $t + m$. At this latter step, the algorithm receives the cost $f_{t+m}(e_{t+m})$ of $H_{e_{t+m}}$ and uses it to update its distribution $x_{t+m+1}$ over the heuristics. This update is performed using the algorithm $\bar{A}$ receiving as input a loss vector $g_{t+m}^{e_{t+m}}$ defined as follows: $g_{t+m}^{e_{t+m}}(i) = f_{t+m}(e_{t+m})/2D$ if $i = e_{t+m}$ and 0 otherwise. Thanks to the assumption that $f_{t+m} \in [0, 2D]^\ell$, we have $g_{t+m} \in [0, 1]^\ell$. Each exploration step is followed by an exploitation step. See the summary of this learning dynamics in Algorithm 1. With $m = 1$, up to the scaling of the loss function, this dynamics would correspond to the classical algorithm for MAB which performs exploration with probability $\epsilon$ and achieves regret $O(T^{2/3})$ (Slivkins, 2019). Note that our setting requires $m \geq 2$.

---

**Algorithm 1:** Learning dynamics

---

1 **Initialization:**
2 $\beta_0 := 0$, $t := 0$, $X := \emptyset$, $E := \emptyset$
3 $\beta_1, \ldots, \beta_T \sim \text{Bernoulli}(\epsilon)$
4 $e_1, \ldots, e_T \sim U(\{1, \ldots, \ell\})$
5 $x_0$ is chosen by $\bar{A}$
6 **while** $t \leq T$ **do**
7    add $t$ to $X$      // Exploitation step
8    **if** $\beta_t = 1$ **then**
9       $t := t + m$     // Skip $m$ steps
10       add $t$ to $E$   // Exploration step
11       $x_{t+1}$ chosen by $\bar{A}$ after feedback $g_t^{e_t}$
12    $t + +$

---

Now, it is enough to describe how to turn the steps of Algorithm 1 into a solution for the original MTS instance. First, we define $x_{t+1} = x_t$ for any $t \notin E$. At $t = 1$, we sample the exploited heuristic from the distribution $x_1$ and at each update of $x_t$, we switch to a different heuristic with probability $\frac{1}{2}\|x_{t-1} - x_t\|_1$ using the procedure *Round* from Proposition 2.2 in order to ensure that, at each time step $t$, we are following heuristic $i$ with probability $x_t(i)$. The state of the exploited heuristic is not known to us $m$ steps before each exploitation step and another $m$ steps after. During these time steps, we make greedy steps from the last

known state $s_{t'}^{i_{t'}}$ of the heuristic $H_{i_{t'}}$ exploited at time $t'$. Namely, we choose $s_t := \arg\min_{s \in M}(d(s_{t'}^{i_{t'}}, s) + c_t(s))$. This procedure is summarized in Algorithm 2.

---

**Algorithm 2:** Producing solution for MTS

---

1 **Input:** $x_0, \ldots, x_T$ produced by Algorithm 1
2 $i_0 \sim U(\{1, \ldots, \ell\})$
3 **for** $t = 1, \ldots, T$ **do**
4    **if** $x_t \neq x_{t-1}$ **then** $i_t \sim \text{Round}(i_{t-1}, x_{t-1}, x_t)$
5    **else** $i_t := i_{t-1}$
6    **if** $s_t^{i_t}$ *is known* **then**
7       go to state $s_t := s_t^{i_t}$ and set $b_t := s_t$
8    **else** /* $m$ states before and after exploration */
9       set $b_t := b_{t-1}$    /* last successful query */
10       $s_t := \arg\min_{s \in M}(d(b_t, s) + c_t(s))$

---

**Lemma 3.1.** *Given $x_1, \ldots, x_T \in [0, 1]^\ell$ produced by Algorithm 1, the expected cost of Algorithm 2 is at most*

$$\left(1 + O\left(\epsilon m^2\right)\right) \sum_{t=1}^{T} \left(f_t^T x_t + D\|x_{t-1} - x_t\|_1\right).$$

A similar statement is proved in (Antoniadis et al., 2023b). We include our proof in Appendix A.1.

**Choice of hyperparameters.** To achieve the regret bound in Theorem 3.9, we choose the parameter $\epsilon := (D\ell \ln \ell)^{1/3} m^{-4/3} \text{OPT}_{\leq 0}^{-1/3}$ for Algorithm 1 and the learning rate $\eta$ of HEDGE which is used as $\bar{A}$ is chosen based on $\gamma := (D\ell \ln \ell)^{1/3} m^{2/3} \text{OPT}_{\leq 0}^{-1/3}$, where $\gamma = 1 - \exp(-\eta)$. While $D, \ell, m$ are usually known from the problem description, $\text{OPT}_{\leq 0}$ can be guessed by doubling as described in Appendix G.

## 3.3. Analysis

We introduce the following random variables which will be useful in our analysis. We define $X_t$ as an indicator variable such that $X_t = 1$ if $t \in X$ and 0 otherwise. Similarly, $E_t$ is an indicator of $t \in E$. These variables are determined by $\beta_1, \ldots, \beta_T$. We also define $g_t = E_t \cdot g_t^{e_t}$ which is 0 for any $t \notin E$. We consider a filtration $\mathcal{F}_0 \subseteq \mathcal{F}_1 \subseteq \cdots \subseteq \mathcal{F}_T$, where $\mathcal{F}_t$ is a $\sigma$-algebra generated by the realizations of $\beta_1, \ldots, \beta_t$ and $e_1, \ldots, e_t$. Note that these realizations determine $X_{t'}, E_{t'}, g_{t'}$, and $x_{t'+1}$ for any $t' \leq t$. Moreover, we have the following observations.

**Observation 3.2.** *For any $t = 0, \ldots, T - m$, we have $E_{t+m} = \beta_t X_t$.*

This is because $t + m \in E$ if and only if $t \in X$ and $\beta_t = 1$.

**Observation 3.3.** *For any $i = 1, \ldots, m$ and $t = 0, \ldots, T - i$, we have $X_{t+i} \geq \prod_{j=0}^{i-1}(1 - \beta_{t+j})X_t$.*

This holds because if $X_t = 1$ and $\beta_t = \cdots = \beta_{t+i-1} = 0$, then also $X_{t+i} = 1$.

**Observation 3.4.** *For $i = 1, \ldots, m$ and $t = i, \ldots, T$, we have* $\mathbb{E}[g_t \mid \mathcal{F}_{t-i}] = E_t f_t / (2D\ell)$.

This follows from $\mathbb{E}[g_t \mid \mathcal{F}_{t-i}] = \mathbb{E}[E_t g_t^{e_t} \mid \mathcal{F}_{t-i}] = \mathbb{E}[\beta_{t-m} X_{t-m} g_t^{e_t} \mid \mathcal{F}_{t-i}] = E_t \cdot \mathbb{E}[g_t^{e_t} \mid \mathcal{F}_{t-i}]$.

Using Observation 3.4, we estimate the cost of the optimal solution $\mathrm{OPT}_{\leq 0}$. Missing proofs are in Appendix A.

**Lemma 3.5.** *Let $x^* \in \Delta^\ell$ be a solution minimizing $\sum_{t=1}^T f_t^T x^*$. We have*

$$\mathbb{E}\left[\sum_{t=1}^T g_t^T x^*\right] \leq \frac{\epsilon}{2D\ell} \mathrm{OPT}_{\leq 0}.$$

The next lemma will be used to bound the costs perceived by Algorithm 1 at time steps $t$ such that $X_{t-m} = 1$.

**Lemma 3.6.** *For each $i = 0, \ldots, m$, we have*

$$\mathbb{E}\left[\sum_{t=m+1}^T g_t^T x_t\right] = \frac{\epsilon}{2D\ell} \mathbb{E}\left[\sum_{t=m+1}^T f_t^T x_{t-i} X_{t-m}\right].$$

Bounding costs in time steps $t$ when $X_{t-m} = 0$ is more involving. In such case, there is $E_{t-i} = 1$ for some $i \in \{1, \ldots, m\}$. We will use the following stability property.

**Lemma 3.7.** *If $\bar{A}$ satisfies Property 2.3, the following holds for every $t \geq m + 1$ and $i \in \{1, \ldots, m\}$:*

$$\mathbb{E}[f_t^T x_t E_{t-i}] \leq \mathbb{E}[f_t^T x_{t-i} E_{t-i}] + \frac{\eta}{\ell} \mathbb{E}[f_{t-i}^T x_{t-i} E_{t-i}].$$

The following lemma is the core of our argument. In the proof, it decomposes the costs perceived by Algorithm 1 in each step $t$ depending on the value of $X_{t-m}$. If $X_{t-m} = 1$, such costs are easy to bound using Lemma 3.6. Those with $X_{t-m} = 0$ need to be charged to some other exploitation step using the preceding stability lemma.

**Lemma 3.8.** *If $\bar{A}$ satisfies Property 2.3 then $\mathbb{E}\left[\sum_{t=1}^T f_t^T x_t\right]$ is at most*

$$2m + \left(1 + \frac{m\epsilon}{(1-\epsilon)^m} + \frac{m\eta\epsilon}{\ell}\right) \cdot \frac{2D\ell}{\epsilon} \mathbb{E}\left[\sum_{t=1}^T g_t^T x_t\right].$$

*Proof.* We start with an observation that for any $t \geq m + 1$, exactly one of the following holds: either $X_{t-m} = 1$ or the step $t - m$ is skipped due to the exploration at time $t - i$ (i.e., $E_{t-i} = 1$) for some $i \in 1, \ldots, m$. We can write

$$\mathbb{E}\left[\sum_{t=1}^T f_t^T x_t\right] \leq 2m + \mathbb{E}\left[\sum_{t=2m+1}^T f_t^T x_t \cdot X_{t-m}\right]$$
$$+ \sum_{i=1}^m \mathbb{E}\left[\sum_{t=2m+1}^T f_t^T x_t \cdot E_{t-i}\right].$$

By Lemma 3.6, the first expectation in the right-hand side is at most $\frac{2D\ell}{\epsilon} \mathbb{E}[\sum_{t=1}^T g_t^T x_t]$. In order to prove the lemma, it is therefore enough to show that, for each $i = 1, \ldots, m$, $\mathbb{E}\left[\sum_{t=2m+1}^T f_t^T x_t \cdot E_{t-i}\right]$ is at most

$$\left(\frac{\epsilon}{(1-\epsilon)^m} + \frac{\eta\epsilon}{\ell}\right) \cdot \frac{2D\ell}{\epsilon} \mathbb{E}\left[\sum_{t=1}^T g_t^T x_t\right]. \quad (1)$$

First, we use Lemma 3.7 to obtain

$$\sum_{t=2m+1}^T \mathbb{E}[f_t^T x_t E_{t-i}] \leq \sum_{t=2m+1}^T \mathbb{E}[f_t^T x_{t-i} E_{t-i}] \quad (2)$$
$$+ \frac{\eta}{\ell} \sum_{t=2m+1}^T \mathbb{E}[f_{t-i}^T x_{t-i} E_{t-i}].$$

The second term is easy to bound: by Observation 3.2, we have $\mathbb{E}[f_{t-i}^T x_{t-i} E_{t-i}] = \mathbb{E}[\beta_{t-i-m}] \mathbb{E}[f_{t-i}^T x_{t-i} X_{t-i-m}]$. Therefore, we can write

$$\frac{\eta}{\ell} \sum_{t=2m+1}^T \mathbb{E}[f_{t-i}^T x_{t-i} E_{t-i}] \leq \frac{\eta\epsilon}{\ell} \sum_{t=m+1}^T \mathbb{E}[f_t^T x_t X_{t-m}].$$
$$(3)$$

Using Observation 3.2, independence of $X_{t-i-m}$ and $\beta_{t-j-m}$ for $j = 1, \ldots, i$, and Observation 3.3, we get

$$\sum_{t=2m+1}^T \mathbb{E}[f_t^T x_{t-i} E_{t-i}] (1-\epsilon)^i$$
$$= \sum_{t=2m+1}^T \mathbb{E}[\beta_{t-i-m}] \mathbb{E}[f_t^T x_{t-i} X_{t-i-m}] \mathbb{E}\left[\prod_{j=1}^i (1-\beta_{t-j-m})\right]$$
$$\leq \sum_{t=2m+1}^T \mathbb{E}[\beta_{t-i-m}] \mathbb{E}[f_t^T x_{t-i} X_{t-m}].$$

In other words, we can bound the first term of (2) as

$$\sum_{t=2m+1}^T \mathbb{E}[f_t^T x_{t-i} E_{t-i}] \leq \frac{\epsilon}{(1-\epsilon)^i} \sum_{t=2m+1}^T \mathbb{E}[f_t^T x_{t-i} X_{t-m}].$$
$$(4)$$

Now it is enough to apply Lemma 3.6 to the right-hand sides of (4) and (3). Equation (2) then implies (1) which concludes the proof. $\square$

**Theorem 3.9.** *Algorithm 2 using HEDGE as $\bar{A}$ with $m$-delayed bandit access to $\ell$ heuristics on any MTS input instance with diameter $D$ such that $D\ell m \leq o(\mathrm{OPT}_{\leq 0}^{1/3})$ satisfies the following regret bound:*

$$\mathbb{E}[\mathrm{ALG}] \leq \mathrm{OPT}_{\leq 0} + O\left((D\ell \ln \ell)^{1/3} m^{2/3} \mathrm{OPT}_{\leq 0}^{2/3}\right).$$

*Proof.* By Lemma 3.1, $\mathbb{E}[\text{ALG}]$ is at most

$$(1 + O(m^2\epsilon))\left(\sum_{t=1}^{T}\mathbb{E}[f_t^T x_t] + \sum_{t=1}^{T}\mathbb{E}[D\|x_{t-1} - x_t\|_1]\right).$$

The first term in the parenthesis can be bounded using Lemma 3.8. The second term can be bounded using Property 2.3: denoting $\eta$ the learning rate of $\bar{A}$, we have

$$\sum_{t=1}^{T}\mathbb{E}[D\|x_{t-1} - x_t\|_1] \le D\eta\sum_{t=1}^{T}\mathbb{E}[g_t^T x_t].$$

Altogether, using $(1 - \epsilon)^m > 1/2$ and $D\eta \le m\epsilon \cdot \eta \cdot \frac{2D\ell}{\epsilon}$, we get that $\mathbb{E}[\text{ALG}]$ is at most

$$\left(1 + O(m^2\epsilon)\right)\left(2m + \left(1 + O(m\epsilon)(1 + \eta)\right)\frac{2D\ell}{\epsilon}\sum_{t=1}^{T}\mathbb{E}[g_t^T x_t]\right).$$

Now we use the regret bound of $\bar{A}$ (Proposition 2.4). Let $H_{i^*}$ be the best heuristic and $x^* \in [0,1]^\ell$ be the vector such that $x^*(i^*) = 1$ and $x^*(i) = 0$ for every $i \ne i^*$. We have

$$\frac{2D\ell}{\epsilon}\sum_{t=1}^{T}\mathbb{E}[g_t^T x_t] \le (1 + \gamma)\frac{2D\ell}{\epsilon}\sum_{t=1}^{T}\mathbb{E}[g_t^T x^*] + \frac{2D\ell \ln \ell}{\epsilon\gamma}$$

$$\le (1 + \gamma)\,\text{OPT}_{\le 0} + \frac{2D\ell \ln \ell}{\epsilon\gamma},$$

where the second inequality used Lemma C.2. In total, $\mathbb{E}[\text{ALG}]$ is at most

$$\text{OPT}_{\le 0} + O(m^2\epsilon + \gamma + m^2\epsilon\gamma)\,\text{OPT}_{\le 0} + O\left(\frac{2D\ell \ln \ell}{\epsilon\gamma}\right).$$

It is enough to choose $\epsilon := (D\ell \ln \ell)^{1/3}m^{-4/3}\,\text{OPT}_{\le 0}^{-1/3}$ and $\gamma := (D\ell \ln \ell)^{1/3}m^{2/3}\,\text{OPT}_{\le 0}^{-1/3}$ to get the desired bound. □

**Theorem 3.10.** *Algorithm 2 with SHARE as $\bar{A}$ and $m$-delayed bandit access to $\ell$ heuristics on any MTS instance with diameter $D$ with offline optimum cost at least $2k$ such that $D\ell m \le o(\text{OPT}_{\le k}^{1/3})$ achieves $\text{Reg}_k(\text{ALG})$ at most*

$$O\left((D\ell k)^{1/3}m^{2/3}\,\text{OPT}_{\le k}^{2/3}\ln\frac{\ell^{1/3}(\text{OPT}_{\le k})^{2/3}}{(Dk)^{2/3}m^{4/3}}\right).$$

The proof can be found in Appendix C.

## 4. Tight Lower Bound: Proof of Theorem 1.3

In this section, we prove a lower bound matching Theorem 3.9. Our proof is based on the construction of Dekel et al. (2013) for Bandits with Switching Costs, which is a special case of Bandit Learning against a 1-Memory Bounded Adversary. In order to use their construction in our setting, we need to address additional challenges due to the algorithm having more power in our setting:

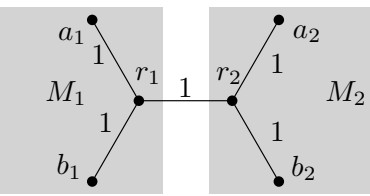

*Figure 1.* With $\ell = 2$, $M$ is the metric closure of this graph.

1. The algorithm can see the full MTS input instance as it arrives online, and therefore it can take actions on its own regardless of the advice of the queried heuristic.

2. The algorithm (as it is common in MTS setting) has 1-step lookahead, i.e., it observes the cost function $c_t$ first and only then chooses the state $s_t$.

Note that both properties are indispensable in our model, see Section 2, and any non-trivial positive result would be impossible without them.

We create an MTS input instance circumventing the advantages of the algorithm mentioned above. Our instance is generated at random and is oblivious to the construction of Dekel et al. (2013). However, we make a correspondence between the timeline in our MTS instance and in their Bandit instance: our instance consists of blocks of length three, where each block corresponds to a single time step in their Bandit instance. The construction of Dekel et al. (2013) itself comes into play when generating the solutions of the heuristics. We create these solutions in such a way that the cost of the heuristic $H_i$ can be related to the cost of the $i$th arm in (Dekel et al., 2013). Moreover, the state of $H_i$ depends only on the loss of the $i$th arm at given time.

**Proposition 4.1** (Dekel et al. (2013))**.** *There is a stochastic instance $\ell_1, \dots, \ell_T$ of Bandits with Switching Costs such that the expected regret of any deterministic algorithm producing solution $i_1, \dots, i_T \in [\ell]$ is*

$$\mathbb{E}\left[\sum_{t=1}^{T}(\ell_t(i_t) + \mathbb{1}(i_t \ne i_{t-1}))\right] - \min_{i \in [\ell]}\sum_{t=1}^{T}\ell_t(i) \ge \tilde{\Omega}(\ell^{\frac{1}{3}}T^{\frac{2}{3}}).$$

**Description of the MTS instance.** The metric space $M$ consists of $\ell$ parts $M_1, \dots, M_\ell$, where $M_i = \{r_i, a_i, b_i\}$. The distances are chosen as follows: For $j \ne i$, we have $d(r_i, r_j) = 1$, $d(r_i, a_j) = d(r_i, b_j) = 2$, and $d(r_i, a_i) = d(r_i, b_i) = 1$. We have $d(a_i, b_j) = 2$ if $i = j$ and 3 otherwise. See Figure 1 for an illustration.

The input sequence consists of blocks of length three. We choose the cost functions in block $j$ as follows. For $i = 1, \dots, \ell$, we choose $\sigma_j^i \in \{a_i, b_i\}$ uniformly at random. For the first step of the block $j$, we define the cost function $c_j'(s)$ such that $c_j'(s) = 0$ if $s \in \{r_1, \dots, r_\ell\}$ and $c_j'(s) = +\infty$

otherwise. In the second time step, we issue $c''_j(s) = +\infty$ if $s \in \{r_1, \ldots, r_\ell\}$ and $c''_j(s) = 0$ otherwise. In the third time step, we issue $c'''_j(s) = 0$ if $s \in \{\sigma^1_j, \ldots, \sigma^\ell_j\}$ and $c'''_j(s) = +\infty$ otherwise[4].

Here is the intuition behind this construction. During block $j$, any reasonable algorithm stays in $s' \in \{r_1, \ldots, r_\ell\}$ in the first step and in $s''' \in \{\sigma^1_j, \ldots, \sigma^\ell_j\}$ in the third step. Ideally, the algorithm would move to $s'''$ already in the second step. However, it does not yet know $\sigma^1_j, \ldots, \sigma^\ell_j$ and therefore needs the advice of the heuristics. In the first step of each block, the algorithm pays 1 if it is staying in the same part $M_i$ of the metric space (for returning to $r_i$). If it is moving from $M_j$ to $M_i$, it has to pay 2, i.e., an additional unit cost.

**Description of the heuristics.** For each $i$, the heuristic $H_i$ remains in $M_i$ throughout the whole input instance. In the first step of the block $t$, it moves to the state $r_i$. In the third step, it moves to $\sigma^i_t$. Its position in the second step is derived from the Bandit instance. With probability $(1 - \frac{\ell_t(i)}{2})$, it is $\sigma^i_t$, and with probability $\frac{\ell_t(i)}{2}$ it is the other point, i.e., $\{a_i, b_i\} \setminus \{\sigma^i_t\}$. In the block $t$, the heuristic $H_i$ pays 2 for the movements in the steps 1 and 2 and, with probability $\frac{\ell_t(i)}{2}$, another 2 for the movement in the step 3. Therefore, we have the following observation.

**Observation 4.2.** *The expected cost of the heuristic $H_i$ is equal to* $2T + \sum_{t=1}^T \ell_t(i)$.

We consider the following special class of algorithms for our MTS instance called *tracking* algorithms. This class has two important properties: firstly, any algorithm on our MTS instance can easily be converted to a tracking algorithm without increasing its cost. Secondly, solutions produced by tracking algorithms can be naturally converted online to the problem of Bandits with Switching Cost.

**Definition 4.3.** Consider an algorithm $A$ with a bandit access to heuristics $H_1, \ldots, H_\ell$. We say that $A$ is a *tracking algorithm* if, while processing the MTS instance described above, satisfies the following condition in every block. Let $H_i$ be the heuristic queried in the second step. Then the algorithm resides in $M_i$ during the whole block and moves to the state of $H_i$ in its second step.

**Lemma 4.4.** *Any algorithm $A$ with bandit access to $H_1, \ldots, H_\ell$ can be converted to a tracking algorithm $\bar{A}$ without increasing its expected cost.*

Proof can be found in Appendix B.

**Lemma 4.5.** *Consider a tracking algorithm $A$. For each block $t$, define $i_t$ such that $A$ is located at $s^{i_t}_t$ in its second step. The expected cost of $A$ is $2T + \sum_{t=1}^T \ell_t(i_t) +$*

$\sum_{t=1}^T \mathbb{1}(i_t \neq i_{t-1})$.

*Proof.* In each block $t = 1, \ldots, T$, the algorithm pays $2 + \mathbb{1}(i_t \neq i_{t-1})$ in the first two steps. In the third step, it pays 2 only if $s^{i_t}_t \neq \sigma^{i_t}_t$ which happens with probability $\frac{\ell_t(i_t)}{2}$. $\qquad\square$

Theorem 1.3 is derived from the following statement by Yao's principle and from the fact that we can pad the constructed instance with zero cost vectors to get an input instance of length higher than $\mathrm{OPT}_{\leq 0}$.

**Theorem 4.6.** *There is a stochastic instance $I$ of MTS of length $3T$ with heuristics $H_1, \ldots, H_\ell$ such that any deterministic algorithm with bandit access to $H_1, \ldots, H_\ell$ suffers expected regret at least $\tilde\Omega(\ell^{1/3}T^{2/3})$.*

*Proof.* Let $I$ be the input instance constructed above from $\ell_1, \ldots, \ell_T$ in Proposition 4.1. Consider a fixed deterministic algorithm $A$. Firstly, we consider queries made by $A$. The queries in the first and the third step of each block have a trivial answer which are independent of the loss sequence $\ell_1, \ldots, \ell_T$: if $A$ queries $H_i$, then the answers are $r_i$ and $\sigma^i_t$ respectively. For each block $t = 1, \ldots, T$, we denote $i_t$ the heuristic queried in its second step. Note that the result of the query depends only on $\ell_t(i_t)$.

We can assume that $A$ is tracking, since making it tracking would only decrease its expected cost. Therefore, the expected cost of $A$ can be written as

$$\sum_{t=1}^T \mathbb{E}\big[\mathbb{1}(i_t \neq i_{t-1}) + 2 + 2 \cdot \mathbb{1}(s^{i_t}_t \neq \sigma^i_t)\big]$$

$$= 2T + \sum_{t=1}^T \mathbb{E}\left[\mathbb{1}(i_t \neq i_{t-1}) + 2\mathbb{E}[\mathbb{1}(s^{i_t}_t \neq \sigma^i_t) \mid \ell_t(i_t)]\right]$$

$$= 2T + \sum_{t=1}^T \mathbb{E}[\mathbb{1}(i_t \neq i_{t-1}) + \ell_t(i_t)].$$

Let $H_{i^*}$ denote the best heuristic. Using Observation 4.2, the regret of $A$ with respect to $H_{i^*}$ is equal to

$$\sum_{t=1}^T \mathbb{E}[\mathbb{1}(i_t \neq i_{t-1}) + \ell_t(i_t)] - \sum_{t=1}^T \mathbb{E}[\ell_t(i^*)].$$

This is equal to the expected regret of the strategy playing arms $i_1, \ldots, i_T$ on the sequence $\ell_1, \ldots, \ell_T$. By Proposition 4.1, this regret is at least $\tilde\Omega(\ell^{1/3}T^{2/3})$. $\qquad\square$

## Acknowlegements

This research was supported by Junior Researchers' Grant which was awarded by Bocconi University thanks to the philanthropic gift of the Fondazione Romeo ed Enrica Invernizzi.

---

[4]We use infinite costs for a cleaner presentation. But choosing $2D = 6$ instead of $+\infty$ does the same job, see discussion in Section 2.

## Impact Statement

This paper presents work whose goal is to advance the field of Machine Learning. There are many potential societal consequences of our work, none of which we feel must be specifically highlighted here.

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

# A. Omitted Proofs from Section 3

## A.1. Proof of Lemma 3.1

We compare the cost of Algorithm 2 to the cost of a hypothetical algorithm $A'$ which, at each time step $t$, is located at state $s_t^{i_t}$. At time step $t$, this algorithm pays cost $C_t' = c_t(s_t^{i_t}) + d(s_{t-1}^{i_{t-1}}, s_t^{i_t})$.

As a proxy for the cost of Algorithm 2, we define $C_t$ in the following way. If $s_t = s_t^{i_t}$, we set $C_t := C_t'$. Otherwise, we set

$$C_t := d(s_{t-1}^{i_{t-1}}, s_t) + c_t(s_t) + d(s_t, b_t) + d(b_t, s_t^{i_t}).$$

This way, the total cost of Algorithm 2 is at most $\sum_{t=1}^{T} C_t$, since the cost of its movement $d(b_t, s_{t+1}) \leq d(b_t, s_t^{i_t}) + d(s_t^{i_t}, s_{t+1})$ at time $t+1$ is split into $C_t$ and $C_{t+1}$.

The following statement is part of the proof of Lemma 5.4 in (Antoniadis et al., 2023b), we include it here for completeness.

**Proposition A.1** (Antoniadis et al. (2023b)). *Consider $t \in \{1, \ldots, T\}$ and denote $\tau \leq t$ the last step such that $s_\tau = s_\tau^{i_\tau}$. Then $C_t \leq C_t' + O(1) \sum_{t'=\tau+1}^{t} C_{t'}'$.*

*Proof.* If $\tau = t$, i.e., $s_t = s_t^{i_t}$, then we have $C_t = C_t'$ and the sum $\sum_{t'=\tau+1}^{t} C_{t'}'$ is empty. Otherwise, $\tau < t$ and the following relations hold due to $b_t = b_\tau = s_\tau^{i_\tau}$, triangle inequality, and the greedy choice of $s_t$ in Algorithm 2 respectively.

$$d(b_t, s_t^{i_t}) = d(b_\tau, s_t^{i_t})$$
$$d(s_{t-1}^{i_{t-1}}, s_t) \leq d(s_{t-1}^{i_{t-1}}, b_t) + d(b_t, s_t)$$
$$d(b_t, s_t) + c_t(s_t) \leq d(b_t, s_t^{i_t}) + c_t(s_t^{i_t}).$$

Plugging this in the definition of $C_t$, we get

$$
\begin{aligned}
C_t &= d(s_{t-1}^{i_{t-1}}, s_t) + c_t(s_t) + d(s_t, b_t) + d(b_t, s_t^{i_t}) \\
&\leq d(b_t, s_{t-1}^{i_{t-1}}) + d(b_t, s_t) + c_t(s_t) + d(s_t, b_t) + d(b_t, s_t^{i_t}) \\
&\leq d(b_t, s_{t-1}^{i_{t-1}}) + 2\big(d(b_t, s_t) + c_t(s_t)\big) + d(b_t, s_t^{i_t}) \\
&\leq [d(b_t, s_{t-1}^{i_{t-1}})] + 2[d(b_t, s_t^{i_t}) + c_t(s_t^{i_t})] + [d(b_t, s_t^{i_t})].
\end{aligned}
$$

Between time steps $\tau$ and $t$, $A'$ has to traverse from $b_t = s_\tau^{i_\tau}$ to $s_{t-1}^{i_{t-1}}$ as well as $s_t^{i_t}$ and pay $c_t(s_t^{i_t})$. Therefore, each bracket in the equation above is bounded by $\sum_{t'=\tau+1}^{t} C_{t'}'$. $\qquad\square$

*Proof of Lemma 3.1.* By Proposition A.1, the cost of Algorithm 2 is at most

$$\sum_{t=1}^{T} C_t \leq \sum_{t=1}^{T} C_t' + O(1) \sum_{t=1}^{T} \sum_{\tau_t+1}^{t} C_t' = \sum_{t=1}^{T} C_t' + O(1) \sum_{t=1}^{T} (a_t - t) C_t',$$

where $\tau_t := \max\{t' \mid t \leq t \text{ and } s_{t'} = s_{t'}^{i_{t'}}\}$ and $a_t := \min\{t' \mid t \geq t \text{ and } s_{t'} = s_{t'}^{i_{t'}}\}$. Therefore, it is enough to show that

$$\mathbb{E}\left[\sum_{t=1}^{T} (a_t - t) C_t'\right] \leq O(\epsilon m^2) \mathbb{E}\left[\sum_{t=1}^{T} C_t'\right].$$

If $a_t > t$, then $s_t \neq s_t^{i_t}$ and there must be an exploration step within $m$ time steps before or after $t$. Therefore, we have

$$
\begin{aligned}
\mathbb{E}\left[(a_t - t) C_t'\right] &\leq \sum_{t'=t-m}^{t+m} \mathbb{E}\left[C_t' E_{t'}(a_t - t)\right] \\
&\leq \sum_{t'=t-m}^{t+m} \mathbb{E}\left[C_t' \mathbb{E}\left[E_{t'}(a_t - t) \mid \mathcal{F}_t\right]\right].
\end{aligned}
$$

Here, we used that $i_t$ and therefore $C_t'$ is determined only by random choices up to time $t$. Now, for each $t' = t - m, \ldots, t + m$, we have

$$\mathbb{E}[E_{t'}(a_t - t) \mid \mathcal{F}_t] = \mathbb{E}[\beta_{t'-m} X_{t'-m}(a_t - t) \mid \mathcal{F}_t] \le \beta_{t'-m}\left(1 + \sum_{i=1}^{T} mi(2m\epsilon)^i\right) \le m\beta_{t'-m}O(1),$$

because in order to have $a_t - t = mi$, we need to have $\beta_{t''} = 1$ in at least once in every block of $2m$ time steps between $t$ to $t + mi$ and $\epsilon < 1/2$. Therefore, we have

$$\mathbb{E}\left[\sum_{t=1}^{T}(a_t - t)C_t'\right] \le O(\epsilon m^2)\mathbb{E}\left[\sum_{t=1}^{T} C_t'\right]$$

which concludes the proof. $\qquad\square$

### A.2. Proofs of Technical Lemmas

*Proof of Lemma 3.5.* By Observation 3.4, we have

$$\mathbb{E}[g_t^T x^*] = \mathbb{E}\left[\mathbb{E}\left[g_t^T x^* \mid \mathcal{F}_{t-1}\right]\right] = \mathbb{E}\left[\frac{E_t}{2D\ell} f_t^T x^*\right]$$

$$= \frac{\mathbb{E}[\beta_{t-m}]}{2D\ell}\mathbb{E}[X_{t-m} f_t^T x^*] \le \frac{\epsilon}{2D\ell} f_t^T x^*$$

for each $t = 1, \ldots, T$. Summing over $t$, we get

$$\mathbb{E}\left[\sum_{t=1}^{T} g_t^T x^*\right] \le \frac{\epsilon}{2D\ell}\sum_{t=1}^{T} f_t^T x^* = \frac{\epsilon}{2D\ell}\,\mathrm{OPT}_{\le 0}. \qquad\square$$

*Proof of Lemma 3.6.* For $t = m + 1, \ldots, T$, we have

$$\mathbb{E}[g_t^T x_t] = \mathbb{E}\left[\mathbb{E}\left[g_t^T \mid \mathcal{F}_{t-1}\right] x_t\right] = \mathbb{E}\left[\frac{E_t}{2D\ell} f_t^T x_t\right]$$

$$= \frac{\mathbb{E}[\beta_{t-m}]}{2D\ell}\mathbb{E}[f_t^T x_t X_{t-m}],$$

where the second equality follows from Observation 3.4. To finish the proof, it is enough to note that $\mathbb{E}[\beta_{t-m}] = \epsilon$ and $x_t X_{t-m} = x_{t-i} X_{t-m}$ for any $i = 0, \ldots, m$. $\qquad\square$

*Proof of Lemma 3.7.* We use the Cauchy-Schwarz inequality and Property 2.3:

$$\mathbb{E}[f_t^T x_t E_{-i}] = \mathbb{E}[(f_t^T(x_t - x_{t-i}) + f_t^T x_{t-i})E_{t-i}]$$

$$\le E[(\|f_t\|_\infty \|x_t - x_{t-i}\|_1 + f_t^T x_{t-i})E_{t-i}]$$

$$\le \frac{\eta}{\ell}\mathbb{E}[f_{t-i}^T x_{t-i} E_{t-i}] + \mathbb{E}[f_t^T x_{t-i} E_{t-i}].$$

The last inequality follows from $\|f_t\|_\infty \le 2D$ and the following computation. Here, we use $x_{t-i+1} = x_t$ whenever $E_{t-i} = 1$, Property 2.3, $x_{t-i}$ and $E_{t-i}$ depending only on time steps up to $t - i - 1$, and Observation 3.4:

$$\mathbb{E}[\|x_t - x_{t-i}\|_1 \cdot E_{t-i}] = \mathbb{E}[\|x_{t-i+1} - x_{t-i}\|_1 \cdot E_{t-i}]$$

$$\le \mathbb{E}[\eta g_{t-i}^T x_{t-i} E_{t-i}]$$

$$= \mathbb{E}\left[\mathbb{E}\left[\eta g_{t-i}^T \mid \mathcal{F}_{t-i-1}\right] x_{t-i} E_{t-i}\right]$$

$$= \frac{\eta}{2D\ell}\mathbb{E}[f_{t-i}^T x_{t-i} E_{t-i}]. \qquad\square$$

# B. Omitted proofs from Section 4

*Proof of Lemma 4.4.* Consider a block such that the algorithm $A$ moves to another part of $M$ in the second step. The cost functions in the first two steps are deterministic, so we can simulate what part the algorithm would go to in the second step and move there already in the first step for the same cost.

Consider a block such that the algorithm moves to another part of $M$ in the third step, i.e., moves from $s \in \{a_i, b_i\}$ to $s' \in \{a_j, b_j\}$. This costs 3 in the third step and the subsequent move to $r_j$ in the beginning of the next block will cost 1. Instead, staying in $M_i$ costs at most 2 in the third step and moving to $r_j$ in the beginning of the next block will cost 2.

Therefore, we can make sure that the algorithm moves between different parts of $M$ only in the first step of each block.

Consider the step 2 of block $t$ where $A$ queries the state $s_t^i$ of $H_i$. If $A$ moves to $s \in \{a_j, b_j\}$, where $j \neq i$, its expected cost will be at least 1, since $\sigma_t^j$ is chosen from $\{a_j, b_j\}$ uniformly at random.

If $A$ moves to $s \in \{a_i, b_i\} \setminus \{s_t^i\}$, Then the expected cost of $A$ in step 3 will be

$$\left(1 - \frac{\ell_t(i)}{2}\right) 2 = 2 - \ell_t(i) \geq 1 \geq \ell_t(i) = 2\frac{\ell_t(i)}{2},$$

where the right-hand side corresponds to the cost of algorithm $\bar{A}$ which moves to $s_t^i$ instead. □

## B.1. Optimality of the Dependence on $D$ and $k$

We elaborate on the asymptotic dependence on $k$ and $D$ for the lower bound with 2-delayed exploration starting from the construction in Section 4.

The optimality of the dependence on $D$ can be seen from scaling. If we scale the input instance in Theorem 4.6 by a factor $D$, i.e. we multiply all distances in the metric space and all cost vectors by $D$, the cost of any algorithm including $\mathrm{OPT}_{\leq 0}$ will be scaled by $D$. Therefore, the lower bound in Thererom 4.6 becomes

$$D \cdot \mathbb{E}[\mathrm{ALG}] \leq D\,\mathrm{OPT}_{\leq 0} + D\tilde{\Omega}(\ell^{1/3}\,\mathrm{OPT}_{\leq 0}^{2/3}) = \mathrm{OPT}'_{\leq 0} + D^{1/3}\tilde{\Omega}(\ell^{1/3}(\mathrm{OPT}'_{\leq 0})^{2/3}),$$

where $\mathrm{OPT}'_{\leq 0} = \mathrm{OPT}_{\leq 0}$ is the new value after the scaling.

Now, we show tightness of $k$ in Theorem 1.2.

**Theorem B.1.** *There exists a stochastic instance $I$ of MTS of length $3T$ with heuristics $H_1, \ldots, H_\ell$ such that any deterministic algorithm with bandit access to $H_1, \ldots, H_\ell$ suffers expected regret at least $\tilde{\Omega}((k\ell)^{1/3}T^{2/3})$.*

*Proof.* Let ALG be any deterministic algorithm for MTS with bandit access to $H_1, \ldots, H_\ell$. We assume without loss of generality that there exists $\tau \in \mathbb{N}$ multiple of 3 such that $3 \cdot T = k \cdot \tau$. We split the time horizon into $k$ segments of size $\tau$. By Theorem 4.6, for each $j \in [k]$ we have

$$\mathbb{E}\left[\sum_{t=\tau(j-1)+1}^{\tau j} \left(f_t^T x_t + d(x_t, x_{t-1})\right)\right] \geq c(H_j^*) + \Omega((\ell)^{1/3}\tau^{2/3}),$$

where $c(H_j^*)$ represents the cost of the best heuristic in the segment $j$. It follows that

$$\mathbb{E}[\mathrm{ALG}] \geq \sum_{j=1}^{k} c(H_j^*) + \Omega(k\ell^{1/3}\tau^{2/3})$$
$$\geq \mathrm{OPT}_{\leq k} + \Omega((k\ell)^{1/3}T^{2/3}). \qquad \square$$

# C. Upper Bound for $m$-Delayed Bandit Access to Heuristics against $\mathrm{OPT}_{\leq k}$

In what follows, we prove an upper bound for Algorithm 1 using SHARE as $\bar{A}$ against $\mathrm{OPT}_{\leq k}$.

Firstly, SHARE satisfies the following performance bound found in (Cesa-Bianchi & Lugosi, 2006).

**Proposition C.1.** *Consider $x_1, \ldots, x_T \in [0,1]^\ell$ the solution produced by SHARE with learning rate $\eta$, sharing parameter $\alpha$, and denote $\gamma := 1 - \exp(-\eta)$. For any solution $i_1, \ldots, i_T$ such that the number of time steps where $i_{t-1} \neq i_t$ is at most $k$, we have*

$$\sum_{t=1}^{T} g_t^T x_t \leq \frac{\ln \frac{1}{1-\gamma}}{\gamma(1-\alpha)} \sum_{t=1}^{T} g_t(i_t) + k \frac{\ln (\ell/\alpha)}{\gamma(1-\alpha)}.$$

The following is a generalization of Lemma 3.5 that holds for $k \geq 0$.

**Lemma C.2.** *Consider $k \geq 0$. Let $i_1, \ldots, i_T \in [\ell]$ be a solution minimizing $\sum_{t=1}^{T} \left( c_t(s_t^{i_t}) + d(s_{t-1}^{i_{t-1}}, s_t^{i_t}) \right)$ such that $i_{t-1} \neq i_t$ holds in at most $k$ time steps $t$. For each $t$, we define $x_t \in [0,1]^\ell$ such that $x_t(i_t) = 1$ and $x_t(i) = 0$ for each $i \neq i_t$. We have*

$$\mathbb{E}\left[ \sum_{t=1}^{T} g_t^T x_t \right] \leq \frac{\epsilon k}{2\ell} + \frac{\epsilon}{2D\ell} OPT_{\leq k}.$$

*Proof.* By Observation 3.4, we have

$$\mathbb{E}[g_t^T x_t] = \mathbb{E}\left[ \mathbb{E}\left[ g_t^T x_t \mid \mathcal{F}_{t-1} \right] \right] = \mathbb{E}\left[ \frac{E_t}{2D\ell} f_t^T x_t \right]$$

$$= \frac{\epsilon}{2D\ell} \mathbb{E}[X_{t-m} f_t^T x_t] \leq \frac{\epsilon}{2D\ell} f_t^T x_t$$

for each $t = 1, \ldots, T$. Let $\delta_t := d(s_{t-1}^{i_t}, s_t^{i_t}) - d(s_{t-1}^{i_{t-1}}, s_t^{i_t}) \leq D$. Summing over $t$, we get

$$\mathbb{E}\left[ \sum_{t=1}^{T} g_t^T x_t \right] \leq \frac{\epsilon}{2D\ell} \sum_{t=1}^{T} f_t^T x_t$$

$$= \frac{\epsilon}{2D\ell} \sum_{t=1}^{T} \left( c_t(s_t^{i_t}) + d(s_{t-1}^{i_{t-1}}, s_t^{i_t}) + \delta_t \right)$$

$$\leq \frac{\epsilon}{2D\ell} \left( OPT_{\leq k} + kD \right),$$

since $\delta_t > 0$ for at most $k$ time steps. $\qquad\square$

We are now ready to prove the following upper bound.

**Theorem C.3.** *Algorithm 2 with SHARE as $\bar{A}$ and $m$-delayed bandit access to $\ell$ heuristics on any MTS instance with diameter $D$ with offline optimum cost at least $2k$ such that $D\ell m \leq o(OPT_{\leq k}^{1/3})$ achieves $\mathrm{Reg}_k(ALG)$ at most*

$$O\left( (D\ell k)^{1/3} m^{2/3} OPT_{\leq k}^{2/3} \ln \frac{\ell^{1/3}(OPT_{\leq k})^{2/3}}{(Dk)^{2/3} m^{4/3}} \right).$$

*Proof.* Let $H_{i_1^*}, \ldots, H_{i_T^*}$ be the sequence of optimal heuristics such that $i_t^* \neq i_{t+1}^*$ for at most $k$ indices $t \in [T-1]$. Then we define $x_t^* \in [0,1]^\ell$ such that $x_t^*(i_t^*) = 1$ and $x_t^*(i) = 0$ for $i \neq i_t^*$.

Proceeding analogously to the proof of Theorem 3.9 we have

$$\mathbb{E}[ALG] \leq \left( 1 + O(m^2\epsilon) \right) \left( 2m + \left( 1 + O(m\epsilon)(1+\eta) \right) \frac{2D\ell}{\epsilon} \sum_{t=1}^{T} \mathbb{E}[g_t^T x_t] \right).$$

Using the regret bound of $\bar{A}$ (Proposition C.1) followed by Lemma C.2 we have

$$\frac{2D\ell}{\epsilon} \sum_{t=1}^{T} \mathbb{E}[g_t^T x_t] \leq \frac{\ln \frac{1}{1-\gamma}}{\gamma(1-\alpha)} \cdot \frac{2D\ell}{\epsilon} \mathbb{E}\left[ \sum_{t=1}^{T} g_t^T x_t^* \right] + 2D\ell k \frac{\ln (\ell/\alpha)}{\epsilon\gamma(1-\alpha)}$$

$$\leq \frac{\ln \frac{1}{1-\gamma}}{\gamma(1-\alpha)} OPT_{\leq k} + (1 + 2\ell) Dk \frac{\ln (\ell/\alpha)}{\epsilon\gamma(1-\alpha)}.$$

By setting $\alpha := (D\ell k)/(\epsilon \, \mathrm{OPT}_{\leq k})$ and $\gamma := \sqrt{D\ell k/(\epsilon \, \mathrm{OPT}_{\leq k})}$ and considering that $k \leq \mathrm{OPT}_{\leq k}/2$, we obtain

$$(1 + \eta)\frac{2D\ell}{\epsilon} \sum_{t=1}^{T} \mathbb{E}[g_t^T x_t] \leq \mathrm{OPT}_{\leq k} + O\left(\sqrt{\frac{D\ell k \, \mathrm{OPT}_{\leq k}}{\epsilon}} \ln \frac{\epsilon \, \mathrm{OPT}_{\leq k}}{Dk}\right) + O\left(D\ell k \ln \frac{\epsilon \, \mathrm{OPT}_{\leq k}}{Dk}\right).$$

where $\eta = \ln 1/(1-\gamma)$ is the learning rate. It follows that the total expected cost is at most

$$\mathbb{E}[\mathrm{ALG}] \leq \mathrm{OPT}_{\leq k} + O(m^2 \epsilon \, \mathrm{OPT}_{\leq k}) + O\left(\sqrt{\frac{D\ell k \, \mathrm{OPT}_{\leq k}}{\epsilon}} \ln \frac{\epsilon \, \mathrm{OPT}_{\leq k}}{Dk}\right) + O\left(m\epsilon D\ell k \ln \frac{\epsilon \, \mathrm{OPT}_{\leq k}}{Dk}\right).$$

Finally, by setting $\epsilon := (\mathrm{OPT}_{\leq k})^{-1/3}(D\ell k)^{1/3} m^{-4/3}$ we obtain the desired bound. $\qquad\square$

**Corollary C.4.** *Algorithm 2 is $(1 + \epsilon)$-competitive against $\mathrm{OPT}_{\leq k}$ for $k$ as large as*

$$\Omega\left(\frac{\epsilon^3 \, \mathrm{OPT}_{\leq k}}{D\ell m^2 (\ln Z)^3}\right)$$

*where $Z = \ell^{1/3}(OPT_{\leq k})^{2/3}/(D^{2/3} m^{4/3})$.*

## D. Upper Bound in the Setting of Arora et al. (2012)

We demonstrate how to use the solutions $x_1, \ldots, x_T$ produced by Algorithm 1 in order obtain solutions in the setting of MAB against $m$-memory bounded adversaries introduced by Arora et al. (2012). To achieve this, we propose Algorithm 3 and analyze its performance. Note that Algorithm 1 does not require any look-ahead, so it can be used directly in the setting of Arora et al. (2012).

---

**Algorithm 3:** Producing solutions for MAB against $m$-memory bounded adversaries

1  **Input:** $x_1, \ldots, x_T$ resulting from Algorithm 1
2  **for** $t = 1, \ldots, T$ **do**
3      **if** $x_t \neq x_{t-1}$ **then** $i_t \sim \mathrm{Round}(i_{t-1}, x_{t-1}, x_t)$
4      **else** $i_t := i_{t-1}$
5      play $i_t$

---

To obtain a regret bound in this setting, we must relate the total cost incurred by Algorithm 3 to $\mathbb{E}[\sum_{t=1}^{T} f_t^T x_t]$ which we can upper bound using Lemma 3.8.

**Lemma D.1.** *Given $x_1, \ldots, x_T \in [0, 1]^\ell$ produced by Algorithm 1, the expected cost of Algorithm 3 is at most*

$$O(m\epsilon T) + \mathbb{E}\left[\sum_{t=1}^{T} f_t^T x_t\right]$$

*Proof.* We begin the proof by observing that $\mathbb{E}[l_t(i_t) \cdot X_t] \leq \mathbb{E}[f_t^T x_t]$, since the algorithm plays $i_t$ sampled from $x_t$ during exploitation rounds. For the time steps leading up to an exploration round and the exploration step itself we can only say that the cost per round incurred is at most 1. We thus have

$$\mathbb{E}[\mathrm{ALG}] = \mathbb{E}\left[\sum_{t=1}^{T} \ell_t(i_t)\right]$$

$$\leq \mathbb{E}\left[\sum_{t=1}^{T} (1 \cdot (1 - X_t) + \ell_t(i_t) \cdot X_t)\right]$$

$$\leq \mathbb{E}\left[\sum_{t=1}^{T}(1 - X_t)\right] + \mathbb{E}\left[\sum_{t=1}^{T} f_t^T x_t\right]$$

It remains to estimate the first term. By definition of $X_t, E_t, \beta_t$ and Observation 3.2 we have:

$$\mathbb{E}\left[\sum_{t=1}^{T}(1 - X_t)\right] \leq 2m + \sum_{i=0}^{m-1}\mathbb{E}\left[\sum_{t=2m+1}^{T}E_{t-i}\right]$$

$$= 2m + \sum_{i=0}^{m-1}\sum_{t=2m+1}^{T}\mathbb{E}[\beta_{t-i-m}]\cdot\mathbb{E}[X_{t-i-m}]$$

$$\leq 2m + \sum_{i=0}^{m-1}\sum_{t=2m+1}^{T}E[\beta_{t-i-m}]$$

$$\leq O(m\epsilon T). \qquad \square$$

We also need to link the real and perceived costs resulting from following an optimal policy.

**Lemma D.2.** *Let $x^* \in \Delta^\ell$ be a solution minimizing $\sum_{t=1}^{T}\ell_t^T x^*$. We have*

$$\mathbb{E}\left[\sum_{t=1}^{T}g_t^T x^*\right] \leq \frac{\epsilon}{2D\ell}\sum_{t=1}^{T}\ell_t^T x^*$$

*Proof.* By Observation 3.4, we have

$$\mathbb{E}[g_t^T x^*] = \mathbb{E}\left[\mathbb{E}\left[g_t^T x^* \mid \mathcal{F}_{t-1}\right]\right] = \mathbb{E}\left[\frac{E_t}{2D\ell}f_t^T x^*\right]$$

$$= \frac{\epsilon}{2D\ell}\mathbb{E}[X_{t-m}f_t^T x^*] \leq \frac{\epsilon}{2D\ell}f_t^T x^*$$

for each $t = 1, \ldots, T$. The second equality holds because $E_t = 1$ only if the same action was taken for the last $m$ steps. Summing over $t$, we get

$$\mathbb{E}\left[\sum_{t=1}^{T}g_t^T x^*\right] \leq \frac{\epsilon}{2D\ell}\sum_{t=1}^{T}f_t^T x^* = \frac{\epsilon}{2D\ell}\sum_{t=1}^{T}\ell_t^T x^*. \qquad \square$$

Putting everything together, we may now prove the following theorem.

**Theorem D.3.** *Consider $\ell$ available arms and let $m \geq 1$ be the memory bound of the adaptive adversary in the setting of Arora et al. (2012). Then Algorithm 3 achieves the following policy regret bound*

$$O\left((m\ell\ln\ell)^{1/3}T^{2/3}\right)$$

*Proof.* We begin the proof by observing that Lemma 3.8 still holds in the setting of Arora et al. (2012) as it is based solely on the dynamics of Algorithm 1 and the stability assumption on $\bar{A}$. Using Lemma D.1 followed by Lemma 3.8 we have

$$\mathbb{E}[ALG] \leq O(m\epsilon T) + \mathbb{E}\left[\sum_{t=1}^{T}f_t^T x_t\right]$$

$$\leq O(m\epsilon T) + 2m + \left(1 + \frac{m\epsilon}{(1-\epsilon)^m} + \frac{m\eta\epsilon}{\ell}\right)\cdot\frac{2D\ell}{\epsilon}\mathbb{E}\left[\sum_{t=1}^{T}g_t^T x_t\right]$$

Arguing as in the proof of Theorem 3.9, we can use Proposition 2.4 and Lemma D.2 to obtain

$$\frac{2D\ell}{\epsilon}\mathbb{E}\left[\sum_{t=1}^{T}g_t^T x_t\right] \leq (1+\gamma)\sum_{t=1}^{T}\ell_t^T x^* + \frac{2D\ell\ln\ell}{\epsilon\gamma}$$

Since $\sum_{t=1}^{T} \ell_t^T x^* \leq T$, it follows that

$$\mathbb{E}[\text{ALG}] - \sum_{t=1}^{T} \ell_t^T x^* \leq O\left((m\epsilon + \gamma)T\right) + O\left(\frac{D\ell \ln \ell}{\epsilon \gamma}\right).$$

Since the losses in the setting of Arora et al. (2012) are contained in $[0,1]^{\ell}$, we may drop the dependence on $D$. By choosing $\epsilon := (\ell \ln \ell)^{1/3} m^{-2/3} T^{-1/3}$ and $\gamma := (m\ell \ln \ell)^{1/3} T^{-1/3}$ we get the desired bound. □

## E. Rounding Algorithm for MTS

We describe a procedure for rounding the solutions produced by fractional algorithms inspired by Blum & Burch (2000) and use it to prove Proposition 2.1 and Proposition 2.2.

For two distributions $p, p' \in \Delta^N$, $\text{EMD}(p, p') = \sum_{i=1}^{N} \max(0, p(i) - p'(i))$. For every $i, j \in [N]$, let $\tau_{i,j} \geq 0$ represent the mass transferred from $p(i)$ to $p'(j)$ such that $p(i) = \sum_{j=1}^{N} \tau_{i,j}$ and $\text{EMD}(p, p') = \sum_{i \neq j} \tau_{i,j}$. Suppose a fractional algorithm chose $i \in [N]$ at the previous time step with the associated distribution $p$. At the next time step (with a new associated distribution $p'$), the algorithm moves to $i' \in [N]$ with probability $\tau_{i,i'}/p(i)$. This procedure is summarized in Algorithm 4 which we also refer to as *Round*. Note that Algorithm 4 can be applied to distributions over the states of a metric space (i.e., $N = M$) as well as distributions over heuristics/arms (i.e., $N = \ell$). We will not mention $N$ explicitly when calling the procedure as a sub-routine for conciseness.

---

**Algorithm 4:** Round

1 **Input:** $i, p, p', N$

2

3 **for** $j = 1, \ldots, N$ **do**

4      $\tau_{i,j} :=$ mass transferred from $p(i)$ to $p'(j)$

5      set $q(j) := \frac{\tau_{i,j}}{p(i)}$

6 sample $i' \sim q$

---

*Proof of Property 2.1.* Given $p_1, \ldots, p_T \in \Delta^M$, let $s_1, \ldots, s_T$ be the solutions produced after iteratively calling Algorithm 4, i.e. $s_t := \text{Round}(s_{t-1}, p_{t_1}, p_t)$ for every $t \in [T]$. Then for every $t \in [T]$ we have

$$\mathbb{E}[c_t(s_t) + d(s_t, s_{t-1})] = \sum_{i=1}^{|M|} \left( c_t(i) p_t(i) + \sum_{j=1, j \neq i}^{|M|} \tau_{i,j} \right) = c_t^T p_t + \text{EMD}(p_{t-1}, p_t),$$

from which we obtain the desired conclusion by summing over $t = 1, \ldots T$. □

*Proof of Property 2.2.* Given $x_1, \ldots, x_T \in \Delta^{\ell}$, let $i_1, \ldots, i_T$ be the solutions produced after iteratively calling Algorithm 4, i.e. $i_t := \text{Round}(i_{t-1}, x_{t_1}, x_t)$ for every $t \in [T]$ and $i_0$ is selected uniformly at random. We subsequently define $s_t^{i_t}$ as the state predicted by heuristic $H_{i_t}$ at time $t$. It follows that for every $t \in [T]$ we have

$$\mathbb{E}[c_t(s_t^{i_t}) + d(s_t^{i_t}, s_{t-1}^{i_{t-1}})] \leq f_t^T x_t + D \cdot \text{EMD}(x_t, x_{t-1}) = f_t^T x_t + \frac{D}{2} \|x_{t-1} - x_t\|_1,$$

where we argued as in the proof of Property 2.1 and used the fact that the distance between any two states is bounded by the diameter $D$. By summing over $t = 1, \ldots T$ we obtain the desired conclusion. □

## F. Online Learning from Expert Advice

### F.1. Classical Algorithms

In this setting, a learner plays an iterative game against an oblivious adversary for $T$ rounds. At each round $t$, the algorithm chooses among $\ell$ *experts*, incurs the cost associated to its choice, and then observes the losses of all the experts at time $t$. In

this full-feedback model, several successful algorithms have been proposed and demonstrated to achieve strong performance guarantees. For an in-depth analysis of these classical algorithms, we invite the reader to see (Cesa-Bianchi & Lugosi, 2006; Blum & Burch, 2000).

**HEDGE.**    The idea behind this algorithm is simple: associate to each expert a weight and then use an exponential update rule for the weights after observing a new loss function. The weights are normalized to produce a probability distribution over the expert set, from which the subsequent action is sampled. The learning dynamics are summarized in Algorithm 5.

---

**Algorithm 5:** HEDGE

---

**1 Input:** $\eta, \ell, T$
**2** $w_0(i) := 1 \quad \forall i \in [\ell]$
**3 for** $t = 1, \ldots, T$ **do**
**4** $\quad W_t = \sum_{i \in [\ell]} w_t(i)$
**5** $\quad x_t(i) = w_t(i)/W_t \quad \forall i \in [\ell]$
**6** $\quad$ Play $i_t \sim x_t$
**7** $\quad$ Observe $g_t \in [0, 1]^\ell$
**8** $\quad$ Set $w_{t+1}(i) := w_t(i) \exp(-\eta \cdot g_{t-1}(i)) \quad \forall i \in [\ell]$

---

**SHARE.**    This algorithm starts from the same exponential update rule as HEDGE, but introduces an additional term. Given the reduction in the sum of the weights $\Delta$, the SHARE algorithm adds to each weight a fixed fraction $\alpha \cdot \Delta$. The effect of this change is that information is *shared* across experts, which makes the algorithm better at tracking the best moving expert. This allows SHARE to achieve good performance with respect to a benchmark that is a allowed to switch experts a limited number of time. Algorithm 6 summarizes these dynamics.

---

**Algorithm 6:** SHARE

---

**1 Input:** $\eta, \alpha, \ell, T$
**2** $w_0(i) := 1 \quad \forall i \in [\ell]$
**3 for** $t = 1, \ldots, T$ **do**
**4** $\quad W_t = \sum_{i \in [\ell]} w_t(i)$
**5** $\quad x_t(i) = w_t(i)/W_t \quad \forall i \in [\ell]$
**6** $\quad$ Play $i_t \sim x_t$
**7** $\quad$ Observe $g_t \in [0, 1]^\ell$
**8** $\quad$ Compute $\Delta := \sum_{i \in [\ell]} w_t(i)(1 - \exp(-\eta \cdot g_t(i)))$
**9** $\quad$ Set $w_{t+1}(i) := w_t(i) \exp(-\eta \cdot g_{t-1}(i)) + \alpha \cdot \Delta \quad \forall i \in [\ell]$

---

### F.2. Proof of Property 2.3

We consider the weight vector $w_t \in [0, \infty)^\ell$ associated to each expert and the loss vector $g_{t-1} \in [0, 1]^\ell$. The distribution over the experts $x_{t-1} \in [0, 1]^\ell$ is obtained as $x_{t-1}(i) = w_{t-1}(i)/W_{t-1}$, where $W_{t-1} = \sum_{j=1}^\ell w_{t-1}(j)$.

The HEDGE algorithm with parameter $\eta > 0$ uses the following update rule

$$w_t(i) := w_{t-1}(i) \cdot \exp\left(-\eta \cdot g_{t-1}(i)\right) \quad \forall i \in [\ell]. \tag{5}$$

The SHARE algorithm with parameters $\eta > 0$ and $\alpha \in [0, 1/2]$ uses the following update rule

$$w_t(i) := w_{t-1}(i) \cdot \exp\left(-\eta \cdot g_{t-1}(i)\right) + \alpha \cdot \Delta/\ell \quad \forall i \in [\ell] \tag{6}$$

where $\Delta := \sum_{j=1}^l (w_{t-1}(j) - \exp\left(-\eta \cdot g_{t-1}(j)\right)w_{t-1}(j))$.

The following statement is part of the proof of Theorem 10 in (Blum & Burch, 2000), which we include here for completeness.

**Lemma F.1.** *The SHARE algorithm satisfies the following*

$$\sum_{i:x_t(i)<x_{t-1}(i)} (x_{t-1}(i) - x_t(i)) \leq \sum_{i:x_t(i)<x_{t-1}(i)} x_{t-1}(i)(1 - \exp(-\eta \cdot g_{t-1}(i))) \tag{7}$$

*for every $t \in [T]$.*

*Proof.* Using the update rule in (6) we have

$$\sum_{i:x_t(i)<x_{t-1}(i)} (x_{t-1}(i) - x_t(i)) = \sum_{i:x_t(i)<x_{t-1}(i)} \left( \frac{w_{t-1}}{W_{t-1}} - \frac{w_{t-1} \cdot \exp(-\eta \cdot g_{t-1}(i)) + \alpha \cdot \Delta/\ell}{W_t} \right)$$

$$\leq \sum_{i:x_t(i)<x_{t-1}(i)} \left( \frac{w_{t-1}}{W_{t-1}} - \frac{w_{t-1} \cdot \exp(-\eta \cdot g_{t-1}(i))}{W_t} \right)$$

$$\leq \sum_{i:x_t(i)<x_{t-1}(i)} \left( \frac{w_{t-1}}{W_{t-1}} - \frac{w_{t-1} \cdot \exp(-\eta \cdot g_{t-1}(i))}{W_{t-1}} \right)$$

where the last inequality uses the observation that $W_t \leq W_{t-1}$ for all $t \in [T]$. $\square$

*Proof of Property 2.3.* The proof for HEDGE follows immediately from Theorem 3 in (Blum & Burch, 2000). The result for SHARE is obtained by applying F.1, followed by the previous argument, since the right-hand side of (7) contains the update rule in (5). As a consequence, for both HEDGE and SHARE, $\eta$ (i.e., the learning rate parameter) directly appears in the expression of Property 2.3. $\square$

## G. Estimating the Baseline Online

The bounds obtained in Theorem 3.9 and Theorem 3.10 require tuning parameters based on the values of $\mathrm{OPT}_{\leq 0}$ and $\mathrm{OPT}_{\leq k}$, respectively. These values are typically unknown a priori and we need to estimate them online. In particular, our algorithm can observe the MTS input instance and, at each time $t$, compute the value of the offline optimal solution up to time $t$. Let OFF denote the cost of the offline optimal solution to the given input instance. If at least one of the heuristics $H_1, \ldots, H_\ell$ achieves cost at most OFF, then $\mathrm{OFF} \leq \mathrm{OPT}_{\leq k} \leq R\,\mathrm{OFF}$. We use OFF as an estimate of $\mathrm{OPT}_{\leq k}$ in order to tune the parameters of our algorithm and achieve regret bound depending on $R$. Note that we do not need to know $R$ beforehand and we can ensure that $R$ is bounded by including some classical online algorithm which is competitive in the worst case among $H_1, \ldots, H_\ell$.

We use the guess and double trick which resembles the classical problem of estimating the time horizon $T$ in a MAB setting, see (Cesa-Bianchi & Lugosi, 2006; Lattimore & Szepesvari, 2017). Here, one starts with a prior estimate of the time horizon and an instance of some online algorithm whose parameters are tuned based on this estimate. Whenever the estimate becomes smaller than the index of the current iteration, the guess is doubled, a new instance of the online algorithm is created, and its parameters are set according to the new guess. In what follows, we show how to adapt this strategy to our setting. We focus on $\mathrm{OPT}_{\leq 0}$, but a similar argument can be made for $\mathrm{OPT}_{\leq k}$.

Let $\mathrm{ALG}(\omega)$ be Algorithm 2 configured with parameters $\epsilon := (D\ell \ln \ell)^{1/3} m^{-4/3} \omega^{-1/3}$ and $\gamma := (D\ell \ln \ell)^{1/3} m^{2/3} \omega^{-1/3}$. Here $D, \ell, m$ are known and $\omega$ is our estimate of the unknown $\mathrm{OPT}_{\leq 0}$. Our strategy is to instantiate a sequence of algorithms $\mathrm{ALG}_1, \mathrm{ALG}_2, \ldots, \mathrm{ALG}_N$ such that $\mathrm{ALG}_i = \mathrm{ALG}(2^i \omega)$ runs between rounds $a_i$ (included) and $b_i$ (excluded). The optimal heuristic on interval $i$ has cost $\mathrm{OPT}_i \leq 2^i \omega$ by construction. Let $\mathrm{OFF}_i$ be the offline MTS optimal cost computed on the interval between $a_i$ and $b_i$. Note that we can always compute this value at time $t$ after observing the sequence of local cost functions $c_{a_i} \ldots, c_{b_i-1}$. Suppose one of the heuristics is $R$-competitive with high probability w.r.t. the offline MTS optimum. Then it must be that $2^i \omega \leq R\,\mathrm{OFF}_i$ with high probability. This effectively provides a threshold for switching to the next instance without knowledge of the true value of $\mathrm{OPT}_i$. It remains to show that this procedure, summarized in Algorithm 7, guarantees a limited overhead w.r.t. to the bound that requires knowledge of $\mathrm{OPT}_{\leq 0}$.

**Proposition G.1.** *Let $D, \ell, m$ be fixed and assume there exists $\bar{H} \in \{H_1, \ldots, H_\ell\}$ such that $\bar{H}$ is $R$-competitive with high probability. Then Algorithm 7 achieves regret $O(R^{2/3} \mathrm{OPT}_{\leq 0}^{2/3})$ with respect to $\mathrm{OPT}_{\leq 0}$.*

---

**Algorithm 7:** Doubling algorithm

---

**1 Input:** $\omega$

**2**

**3** $i = 0$

**4** $t = 1$

**5** $\mathrm{ALG}_0 := \mathrm{ALG}(\omega)$

**6 while** $t \leq T$ **do**

**7**     **if** $\mathrm{OFF}_i > R \cdot 2^i \omega$ **then**

**8**         $i{+}{+}$

**9**         $\mathrm{ALG}_i := \mathrm{ALG}(2^i \omega)$

**10**     Choose $s_t$ according to $\mathrm{ALG}_i$

**11**     $t{+}{+}$

---

*Proof.* By construction of the sequence of algorithms, we have that $2^i\omega \leq \mathrm{OPT}_i$ and $\sum_{i=1}^N \mathrm{OPT}_i \leq \mathrm{OPT}_{\leq 0}$. It follows that

$$\sum_{i=1}^N 2^i\omega = \omega(2^N - 1) \leq OPT_{\leq 0},$$

which implies that $N \leq \log(\mathrm{OPT}_{\leq 0})$. Moreover, we have

$$\begin{aligned}
\mathrm{Reg} &= \mathbb{E}[\mathrm{ALG}] - \mathrm{OPT}_{\leq 0} \\
&= \sum_{i=0}^N \left(\mathbb{E}[\mathrm{ALG}_i] - \mathrm{OPT}_i\right) \\
&\leq \sum_{i=0}^N O(\mathrm{OPT}_i^{2/3}) \\
&\leq O\left(\sum_{i=0}^N (R \cdot 2^{i+2}\omega)^{2/3}\right) \\
&\leq O\left(R^{2/3} \mathrm{OPT}_{\leq 0}^{2/3}\right)
\end{aligned}$$

where the first inequality comes from Theorem 1.1 and the second from the assumption in the hypothesis. $\qquad\square$

The previous result means that we can guarantee regret which is worse by a factor of $R^{2/3}$ w.r.t. to the bound which requires knowledge of $\mathrm{OPT}_{\leq 0}$. $R$ depends on the MTS variant being solved. For general MTS, we can assume $R \leq 2n - 1$ by including the classical deterministic algorithm by Borodin et al. (1992), or $R = O(\log^2 n)$ by including the algorithm by Bubeck et al. (2019). Using the procedure of Komm et al. (2022), any algorithm for MTS which is $R$-competitive in expectation can be made $(1 + \alpha)R$-competitive with high probability for any $\alpha > 0$.

