# OpenReview forum: "Learning-Augmented Algorithms for MTS with Bandit Access to Multiple Predictors"
_ICML.cc/2025/Conference — ICML 2025 poster_

### Official Review · Reviewer_FnST · 2025-02-21

**Overall Recommendation:** 4

**Summary:**

The paper studies the online metrical task system (MTS) problem in a learning-augmented variant. Given a metric space and an initial state $s_0$ (point), at each time a cost function $c_t$ over points is revealed, and an algorithm needs to select a state $s_t$ and pays cost equal to $d(s_{t-1},s_t) + c_t(s_t)$. The goal is to minimize the total cost.

The present paper studies MTS in a setting with expert advice: an online algorithm has access to $\ell$ heuristics, and the goal is to achieve a good performance compared to the best heuristic in hindsight.
Specifically, an algorithm only has bandit access to the heuristics, which means that it does not see the state of each heuristic at every timestep, but can only query the state of at most one heuristic. Moreover, they assume that an algorithm needs to query a heuristic for $m$ consecutive timestep before the current state will be revealed.

The authors present an algorithm with sublinear regret and prove that it is best-possible. Their algorithm reduces the bandit setting to a black box full feedback setting, with a tradeoff between exploration and exploitation. This gives a fraction solution. The authors then present an online rounding scheme.

**Claims And Evidence:**

All claims are proven formally.

**Essential References Not Discussed:**

As far as I see, all relevant literature is being discussed.

**Experimental Designs Or Analyses:**

N/A

**Methods And Evaluation Criteria:**

N/A

**Other Comments Or Suggestions:**

- page 1, line 18 right: replace $x_t$ with $s_t$
- references: there are several arxiv references, for which conference versions exist. I suggest to update them.

**Other Strengths And Weaknesses:**

Strengths
- The paper is well written
- Broadens the discussions on multiple predictions
- Gives tight results, hence answers the research question completely

**Questions For Authors:**

N/A

**Relation To Broader Scientific Literature:**

The key contribution of the paper is the presentation and exhaustive study of a new interesting model within the area of multiple predictions for learning-augmented algorithms. I believe that the results are non-trivial and add valuable insights to this area. Also, the contribution of upper and lower regret bound builds a nice package.

**Theoretical Claims:**

I checked the proofs in the main part of the paper and they seem correct to me.

---

> ### Author Rebuttal · Authors · 2025-03-31
>
> Thank you for your review and for your suggestions. We will incorporate them in
> our manuscript.

---

### Official Review · Reviewer_uNsW · 2025-03-05

**Overall Recommendation:** 4

**Summary:**

The paper studied the learning-augmented metrical task system (MTS) problem and gave near-tight bounds of $\tilde{\Theta}(\text{OPT}^{2/3})$. The problem is similar to the adversarial bandits with $T$ days; however, the ``switching’’ between bandits would involve a cost, which is measured by the distance between two states in a metric space. Let $s_t$ and $s_{t+1}$ be the states (arms) we commit to on time step $t$ and $t+1$, the cost of this step is defined as $c_t(s_t)+d(s_{t}, s_{t+1})$, where $c_t$ is a cost function only known on time step $t$.

In the learning-augmented setting, there are $\ell$ online ML models (known as the ``heuristics’’), and each ML model $i$ would recommend a prediction $s^i_t$ at time $t$. Importantly, we cannot observe the full cost since we do not know what will be the state $s^i_{t+1}$ for model $i$ to switch to in step $t+1$. The goal is to be competitive with the best ML model in hindsight. As standard in the literature, we define the extra additive error as regret.

The main contributions of the paper are as follows:
- An algorithm that achieves $O(\text{OPT}^{2/3})$ regret for the learning-augmented MTS problem (assuming constant $\ell$ and range of the metric space).
- A near-matching lower bound such that any algorithm has to suffer $\tilde{\Omega}(\text{OPT}^{2/3})$ regret.
- The generalization of the results, including results in a setting where the signal delay is $m\geq 2$ and the setting where the algorithm could make at most $k$ switches.

**Claims And Evidence:**

The theoretical results are with full proofs and/or external references.

**Essential References Not Discussed:**

Nothing essential is missing, although I think the paper could expand the discussion on adversarial bandits and learning-augmented algorithms for graph problems.

**Experimental Designs Or Analyses:**

N/A, theoretical paper without experiments.

**Methods And Evaluation Criteria:**

N/A, no experiments included.

**Other Comments Or Suggestions:**

See above.

**Other Strengths And Weaknesses:**

In general, I believe the problem studied in the paper is well-motivated, and the results are interesting. I have worked in both learning-augmented algorithms and online learning, and it is great to see connections between the areas. The analyses are non-trivial, and the paper is written in a relatively clean and coherent manner such that the intuitions are well-explained. The results also demonstrate a nice comparison with ACEPS [ICML’23] and show the importance of the extra information in that work.

On the flip side, I think that although the paper has already made reasonable effort to clarify the problem and the existing work, I still doubt whether people unfamiliar with the literature would find the paper hard to follow. This is partially attributed to the fact that the paper is a follow-up of a long line of existing work. However, maybe the authors should consider an expanded discussion (maybe in the appendix) to help readers less familiar with the literature to understand the problem and the techniques.

**Questions For Authors:**

N/A, I do not have additional questions.

**Relation To Broader Scientific Literature:**

This paper has significance in both machine learning and theoretical computer science. There could also be many applications in online learning.

**Theoretical Claims:**

The main techniques used in this paper are heavily connected to the existing work of ADT [ICML’12] and DDKP [STOC’14]. At a high level, the algorithm tosses a coin to decide whether to follow $m$ steps of exploration or take the next step from an MWU-type of algorithm (e.g., the HEDGE algorithm). Running this algorithm would result in a distribution over the actions at each time step, and we can run a standard rounding algorithm to get the actual action sequence. The bounds on regret intuitively follow from the bounded loss argument in the online learning literature, although there are some additional technical steps for the formal argument. The lower bound was built on the foundation of the lower bound instance in DDKP [STOC’14], and the paper did more work to incorporate the metric switching cost into the argument.

The proofs look intuitively correct to me, although I did not check their correctness in detail.

---

> ### Author Rebuttal · Authors · 2025-03-31
>
> We thank you for your feedback and for highlighting the strengths of our
> contribution. We will incorporate your suggestions in the next revision
> of our manuscript.

---

### Official Review · Reviewer_Jzgp · 2025-03-10

**Overall Recommendation:** 3

**Summary:**

The paper studies the problem of metrical task system (MTS) under the bandit feedback setting. Given multiple heuristic predictors of what action to take, the algorithm can choose one predictor and receive feedback only if the same predictor is used consecutively across m time steps. A tight regret bound was proven and the authors also studied an extension to the setting that one can switch between heuristics at most k times.

**Claims And Evidence:**

Yes

**Essential References Not Discussed:**

Not that I am aware of

**Ethical Review Concerns:**

NIL

**Experimental Designs Or Analyses:**

No experiments as it is a theoretical paper.

**Methods And Evaluation Criteria:**

Yes

**Other Comments Or Suggestions:**

- In Proposition 2.2, you should write "There is an online randomized algorithm Round..." so that it makes sense to refer to the algorithm here as Round subsequently.
- Given that your method uses HEDGE and SHARE as black-boxes, you should provide a description and discussion of them either in the main paper or at least in the appendix.

**Other Strengths And Weaknesses:**

# Strengths
- Asymptotically matching regret bounds were proven (Theorem 1.1 and 1.3) for the problem studied
# Weaknesses
- I do not believe that this work fits into the learning-augmented algorithms framework. It fits much more closely to the bandit or expert selection framework. Typically, a work in this area would require one to prove robustness guarantees showing the performance of the algorithm when the predictor is of arbitrarily poor quality; in this case, one would expect a result showing that the expected cost of ALG degrades back into predictor-free guarantee of $O(\log^2 n)$ when all heuristic predictors are arbitrarily bad. However, the paper only gives bounds with respect to the performance of the best predictor (which could be arbitrarily bad), which is precisely what the bandit or expert selection literature measures against.
- The problem setting of "m-delayed bandit access to heuristics" feels made up just to introduce additional constraints to distinguish itself from the settings of prior works. For instance, I don't see anything algorithmically or theoretically interesting about this m-delay feedback beyond trivially being forced to repeat the same choice m times (see Algorithm 1) and incurring a corresponding factor m in the analyses.

**Questions For Authors:**

- Is there no constraint on number of heuristics $\ell$ and delay length $m$ with respect to the input length? What if we don't even get to execute some heuristic?
- How are the hyperparameters such as $\eta$ in Property 2.3 and $\varepsilon$ in Algorithm 1 determined in practice? How should someone using your algorithm set them? Where does $\eta$ show up in your actual algorithm?
- What are $X$ and $E$ in Algorithm 1? They are undefined and uninitialized, but they should be sets, right?
- Can you point out anything interesting about the algorithms or proof techniques used to obtain your results? Everything looks rather "standard" to me, modulo some minor tweaking to accommodate the m-delay setting (which feels made up to me). I am happy to upgrade my score if I am sufficiently convinced by the rebuttal (or the reviews given by other reviewers) that there is something substantially interesting going on in the paper that I have missed.

**Relation To Broader Scientific Literature:**

This work would be of interest to the bandit and expert selection communities.

**Theoretical Claims:**

I skimmed the proofs. They look reasonable.

---

> ### Author Rebuttal · Authors · 2025-03-31
>
> We provide answers to your questions clarifying the framing of our work
> in the learning-augmented framework and the difficulties present in our setting
> compared to the previous works.
> We believe that these answers could also be valuable to other
> researchers as suggested by
> Reviewer uNsW. We will add a further discussion to our paper.
>
> > I do not believe that this work fits into the learning-augmented algorithms framework... Typically, a work in this area would require one to prove robustness guarantees...
>
> In fact, our result can be used (and is intended) as a tool to robustify any
> learning-augmented algorithm (or any heuristic) with a negligible overhead.
> Let $A_0$ be a classical online algorithm
> and let $A_1$ be a heuristic with no worst-case guarantee.
> On any input sequence, our algorithm is never more than $(1+o(1))$ factor worse
> than the *best* of $A_0$ and $A_1$.
> I.e., if $A_0$ is $R$-competitive for the given MTS variant,
> our algorithm is guaranteed to be at most $(1+o(1))R$-competitive.
> However, if the cost of $A_1$ on the given input is only
> 1.01 times the offline optimum, our algorithm's
> cost will be only factor $(1+o(1))*1.01$ from offline optimum.
>
> Here, it is crucial that our regret guarantees are in terms
> of $OPT$ instead of the time horizon $T$ as common in online learning
> literature, because $OPT$ can be much smaller than $T$ in general MTS inputs.
>
> > The problem setting of "m-delayed bandit access to heuristics" feels made up
>
> Our results, in order to be meaningful in MTS setting, require $m\geq 2$
> consecutive queries to the same heuristic when performing exploration.
> This is necessary to estimate the cost of the
> heuristic: imagine a heuristic which, at each time $t$, moves to a state $s_t$
> such that $c_t(s_t)$ is 0. Typically, this is a very bad idea in MTS, since such
> algorithm would most likely pay very large movement costs.
> However, the movement cost can be calculated only if we know its previous state
> $s_{t-1}$, i.e., if we have queried the same heuristic in the previous time
> step.
>
> Generalization to $m>2$ is inspired by Arora et al.
> In their setting, algorithm has to wait $m$ time steps
> before seeing the relevant feedback. In our case, we do not even see
> the actions taken by the heuristic and we need to find suitable actions on our
> own.
>
> > Can you point out anything interesting... Everything looks rather "standard"
>
> Approach from the previous work of Arora et al.
> would lead to a regret of order $T^{2/3}$ instead of $OPT^{2/3}$,
> see our response to Reviewer syMW, which is not useful for robustification.
> In turn, our algorithm is more similar to the classical algorithm for bandit
> setting alternating exploration and exploitation.
> However, there are three key differences, each of them is necessary to achieve
> our performance guarantees:
>
> * our algorithm makes improper steps (i.e., steps not taken by any of the
> heuristics)
>
> * we use MTS-style rounding to ensure bounded switching cost instead
> of independent random choice at each time step
>
> * exploration steps are not sampled independently since our setting requires $m\geq 2$.
>
> In particular, the last difference leads to much more involving analysis:
> We cannot assume that we have an unbiased estimator of the loss vector
> and therefore we need to do a lot of conditioning on past events.
> Moreover, the cost of only one of the $m>2$ time steps during each exploration can
> be directly charged to the expected loss of the internal full-feedback algorithm
> and the trivial bound of $2D$ is insufficient to achieve regret sublinear in OPT.
> We exploit stability property of the internal full-feedback algorithm
> in order to relate the costs incurred during the $m$ steps of each exploration.
>
> ## Answers to more specific comments:
>
> > Is there no constraint on number of heuristics and delay length with respect to the input length?
>
> In the usual setting of online learning, the number $\ell$ of experts (or
> arms, etc) is fixed while the time horizon $T$ is increasing towards infinity.
> In Section 1, we also state our theorems with $\ell, m, D$ constant which
> we consider the most natural setting.
> However, our bounds hold as far as
> $D\ell m\leq o(OPT^{1/3})$, see Theorems 3.9 and 3.10 for the formal statement.
> Our analysis extends to higher $m$, giving a regret depending on $m^{3/2}$.
>
> > hyperparameters
>
> Optimal choice of the hyperparameters of our algorithms is established at the end
> of the proof of each upper bound (Theorems 3.9 and 3.10).
> Note that $\eta = -\log (1-\gamma)$ in our notation.
> The hyperparameters are chosen based on $D, \ell, m$ which are all known beforehand,
> and OPT, which can be guessed by the standard doubling techniques.
> We will include the details of guessing OPT in our revised manuscript.
>
> > Initialization of $X$ and $E$
>
> Algorithm 1 starts with $X$ and $E$ being empty sets.
>
> We will use your comments to improve our manuscript.

---

> > ### Comment · Reviewer_Jzgp · 2025-04-02
> >
> > Thank you for clarifying my doubts. Your responses are very thoughtful and I am convinced to increase my score. Please add as much of these discussions in a suitable manner into your revision. Thanks!

---

### Official Review · Reviewer_syMW · 2025-03-17

**Overall Recommendation:** 3

**Summary:**

This paper considers the problem of sequentially selecting heuristics for Metrical Task Systems (MTS) when multiple heuristics are available. We focus on the bandit feedback setting, in which only the output of the heuristic chosen at each time step is observable. For this problem, we design and analyze algorithms that minimize the difference in total cost compared to the best fixed heuristic in hindsight, a quantity analogous to regret in bandit problems. Furthermore, we establish lower bounds that nearly match the upper bounds achieved by our proposed algorithms.

**Claims And Evidence:**

The main contributions of this paper are theoretical, and all appear to be supported by correct proofs.

**Essential References Not Discussed:**

For the setting where $ k \geq 1 $, concepts from non-stationary bandits and dynamic regret appear to be relevant, so it would be beneficial to mention them, e.g., Section 8 of [Auer, P., Cesa-Bianchi, N., Freund, Y., & Schapire, R. E. (2002). The nonstochastic multiarmed bandit problem. SIAM journal on computing, 32(1), 48-77.]

**Experimental Designs Or Analyses:**

N/A

**Methods And Evaluation Criteria:**

The evaluation metric used in this paper is natural and reasonable.

**Other Comments Or Suggestions:**

N/A

**Other Strengths And Weaknesses:**

For the proposed algorithm, achieving the guarantees stated in Theorem 3.9 and Theorem 3.10 appears to require setting parameters such as the learning rate based on the value of *OPT*. In other words, prior knowledge of *OPT* or at least an approximation of it seems necessary. However, in practice, *OPT* is often unknown beforehand. This is a potential weakness that is not mentioned in the main text.

Conversely, a strength that is not emphasized in the paper is the dependency on the number of heuristics $ \ell $ in the theoretical bounds. While the introduction does not explicitly discuss this aspect, the derived bounds actually show that both the upper and lower bounds scale approximately as $ \ell^{1/3} $. This means that the guarantees on *OPT* have a tight dependency on $ \ell $, which could be highlighted more prominently.

**Questions For Authors:**

I have a question regarding the connection to *bandits with switching costs* [Dekel et al. (2013)]. This paper uses their results to establish the lower bound, but is it possible to leverage their results for the upper bound as well?

For example, in the case of $ m = 2 $, if we simply define the loss as $ f_t $ or $ g_t $ in your paper and apply their algorithm, wouldn’t we obtain an upper bound of $ (DT)^{2/3} $ instead of $ OPT^{2/3} $?

Additionally, I wonder if some refinement of their analysis could lead to an upper bound of $ OPT^{2/3} $. I would appreciate hearing the authors' thoughts on this point.

**Relation To Broader Scientific Literature:**

This paper considers MTS as an application, while its technical components are primarily based on techniques from online learning and bandit problems. In particular, it appears to be closely related to topics such as bandits with switching costs and non-stationary bandits.

**Theoretical Claims:**

I have briefly checked the proofs of Lemma 3.8, Theorem 3.9, Lemma C.2, and Theorem 4.6.
No particular issues were found.

---

> ### Author Rebuttal · Authors · 2025-03-31
>
> > References Not Discussed
>
> Thank you for the references; we will include them in the revision of our paper.
>
> > In practice, OPT is often unknown beforehand.
>
> We can guess OPT using the doubling technique, getting virtually the same bound, e.g. as in Cesa-Bianchi
> and Lugosi (Section 2.3) or Lattimore, Szepesvari (Section 28.5).
> We thank you for your question. We will add a detailed explanation in the
> revision of our paper.
>
> > Conversely, a strength that is not emphasized is the dependence on the number of heuristics in the theoretical bounds.
>
> Thank you for pointing out this strength of our result. We will make it more
> prominent in the revision.
>
> > question regarding the connection to bandits with switching costs [Dekel et al. (2013)]. is it possible to leverage their results for the upper bound as well?
>
> In MTS, the losses are not a priori bounded. Therefore, we cannot blindly follow
> the advice of the explored heuristic, since its state can have arbitrary large cost.
> However, it is true that if a heuristic suggests a state with a very high
> cost,  we can always find a different (improper) step as described
> in our paper, ensuring that our cost is bounded by $2D$.
> Having this, we can apply
> the algorithm of Arora, Dekel, Tewari (2021)
> for bandits with switching cost and
> achieve a regret of order $O(T^{2/3})$ in our setting.
>
> Their result is formulated as a meta algorithm and proving a formal lower
> bound is not easy.
> However, we do not think that a refinement of their analysis would lead to an
> upper bound in terms of $OPT^{2/3}$. Here is a simple argument:
> Their algorithm
> splits the input sequence into blocks of size $\tau$ and let
> some MAB algorithm choose a single arm (or heuristic) for each block which is then played during the whole block.
> This is to limit the switching cost to $O(T/\tau)$.
> If we want it to be of order $OPT^{2/3}$,
> we need to choose the block size $\tau \geq T/OPT^{2/3}$.
> However, with blocks so large, already a single exploration of some
> very bad heuristic would cause a cost of order $\tau \gg OPT^{2/3}$
> if $OPT$ is small.

---

### Decision · Program_Chairs · 2025-05-01

**Decision:**

Accept (poster)

**Comment:**

Reviewers' scores are all positive, with most agreeing that the problem under study is well-motivated and the results are interesting. Some reviewers initially questioned the naturalness of "m-delayed bandit access to heuristics," but the authors’ response effectively addressed these concerns, convincing the reviewers and leading to scores being increased.